# Ubiquitin-derived artificial binding proteins targeting oncofetal fibronectin reveal scaffold plasticity by β-strand slippage
Anja Katzschmann[1], Ulrich Haupts[1], Anja Reimann[1], Florian Settele[1], Manja Gloser-Bräunig[1], Erik Fiedler [1] ✉ & Christoph Parthier [2] ✉

Affilin proteins, artificial binding proteins based on the ubiquitin scaffold, have been generated by directed protein evolution to yield *de-novo* variants that bind the extra-domain B (EDB) of oncofetal fibronectin, an established marker of tumor neovasculature. The crystal structures of two EDB-specific Affilin variants reveal a striking structural plasticity of the ubiquitin scaffold, characterised by β-strand slippage, leading to different negative register shifts of the β5 strands. This process recruits amino acid residues from β5 towards the N-terminus to an adjacent loop region and subsequent residues into β5, respectively, remodeling the binding interface and leading to target specificity and affinity. Protein backbone alterations resulting from β-strand register shifts, as seen in the ubiquitin fold, can pose additional challenges to protein engineering as structural evidence of these events is still limited and they are difficult to predict. However, they can surface under the selection pressure of directed evolution and suggest that backbone plasticity allowing β-strand slippages can increase structural diversity, enhancing the evolutionary potential of a protein scaffold.

The selection of an appropriate, well-understood structural scaffold typically determines the direction for the development, producibility and potential future applications of artificial binding proteins generated through directed evolution. Tailored for a wide range of applications in medicine, biotechnology and research, an increasing number of artificial binding proteins derived from non-immunoglobulin-based scaffolds have been developed. They exhibit favourable biophysical properties and have the potential to specifically bind to any target molecule[1,2]. Most of these scaffolds are based on naturally occurring, stable and structurally well-characterized protein modules, comprising secondary structural elements such as α-helices, β-sheets and loops to varying degrees. They include lipocalins[3], ankyrin repeat proteins[4], staphylococcal protein A[5], fibronectin[6], γ-crystallin[7], ubiquitin[8,9] and others[10]. These modules are typically chosen as structural platforms or building blocks for the directed evolution of novel binding properties. This is achieved primarily through specific amino acid exchanges at selected sites. Screening of combinatorial scaffold protein libraries using appropriate selection techniques enables the isolation of candidate binders with the desired properties. The success of the random evolution approach is highly dependent on the selection of amino acid

positions to be diversified[11]. Incorporating residue insertions or deletions (InDels) into evolutionary strategies introduces additional complexity, as they may result in larger, non-local structural changes[12]. Scaffold selection and library design usually depend on existing structural knowledge and assume that the scaffold protein's overall fold, including the composition and spatial arrangement of secondary structure elements, is essentially fixed[13].

The structure of ubiquitin (Ub), a 76 amino acid signalling protein conserved among all eukaryotes, features a single five-stranded β-sheet enclosing a short α-helix. Approaches for using a single Ub chain as a scaffold to generate binding proteins[8,14] have been extended through the development of diubiquitin-based scaffolds (Affilin molecules). They are generated by genetic fusion of two Ub domains and have been evolved to target for example oncofetal fibronectin[15]. This fibronectin (Fn) splice variant contains the extra-domain B (EDB) which is specifically expressed during tumour-associated angiogenesis and other neoplastic processes[16,17].

Here, we describe the directed evolution and structural determination of Affilin variants targeting oncofetal Fn, resulting in unexpected structural

[1]Navigo Proteins GmbH, Heinrich-Damerow-Straße 1, 06120 Halle (Saale), Germany. [2]Martin-Luther-University Halle-Wittenberg, Institute of Biochemistry and Biotechnology, Kurt-Mothes-Straße 3a, 06120 Halle (Saale), Germany. ✉e-mail: erik.fiedler@navigo-proteins.com; christoph.parthier@biochemtech.uni-halle.de

rearrangements through different β-strand register shifts, reflecting the plasticity of the diubiquitin scaffold and enabling the formation of the target binding site. Moreover, with a view to potential future applications, we show that a genetic Affilin-cytokine fusion is functional and able to target EDB-expressing cells in vitro.

## Results

### Generation of Affilin variants targeting EDB of oncofetal fibronectin

The development of dibiquitin-based artificial binding proteins with evolved binding properties against human oncofetal Fn has been previously described[15]. Using in silico analysis, potential binding epitopes of the Ub scaffold with a high tolerance for amino acid substitutions were evaluated by assessing protein stability perturbations induced by single amino acid exchanges in wild-type Ub. Nine amino acid positions (2, 4, 6, 8, 62–66, positions 2–8 located in strand β1 and loop β1β2, positions 62–66 in loop α2β5 and strand β5) were selected for randomisation, from which subsets of 8 and 6 positions were used for saturation mutagenesis of the two Ub domains (sequence overview given in Supplementary Fig. S1). We followed a similar approach to generate the Affilin Af2, employing the diubiquitin scaffold with a subset of 7 of the same randomised positions in each Ub domain, a modified library design and advanced selection and maturation procedures. In brief, a library of diubiquitin variants was generated by joining two libraries of monomeric Ub (variant Ub-WAA), each randomised at amino acid positions 6, 8 and 62–66, which was then used to select EDB-binding affilin variants by phage display (PD). The effective library size used in PD was calculated to be $2.5 \cdot 10^9$. Subsequent thermodynamic stability screening selected the affilin variant Af2p, comprising 14 evolved amino acid changes compared to diUb-WAA, including Lys6His, Leu8Asp, Gln62Asp, Lys63Pro, Glu64Gln, Ser65Leu, Thr66Lys, Lys85Thr, Leu87Gln, Lys141Asp, Glu142Tyr, Ser143Arg, Thr144Tyr, and the deletion of Gln140. Af2p displayed nanomolar binding affinity to the target 67B89 and no binding to the Fn fragment 6789, which lacks the EDB. However, the thermal stability decreased significantly ($\Delta T_m = -18K$) compared to the parental variant diUb-WAA (Table 1, Fig. 1a–c).

For affinity maturation of Af2p, we opted for a broader mutagenesis approach by error-prone PCR, allowing re-randomisation of amino acids at any position. Selection of molecules binding to the target 67B89 was performed using four rounds of ribosome display (RD), taking into account a

theoretical library size of $6 \cdot 10^{10}$ variants, which is significantly larger compared to the library used in PD. Stringency of RD was iteratively increased with every cycle to select for binders with a low off-rate. In order to prevent frame-shift variants from leading to the selection of unspecific binders, a screening procedure was implemented in which a C-terminal eGFP fusion is added to the pool of binders, allowing the identification of candidates with the correct reading frame. The final Affilin variant Af2s (produced without the eGFP fusion) was nominated based on its target binding affinity and specificity, thermal stability, and recombinant producibility. Af2s differed from Af2p by having two additional mutations (Pro38Gln and Tyr143Phe) and a randomly arisen deletion (ΔIle78) in the linker region between the N- and C-terminal Ub domains. The maturation process resulted in higher binding affinities for Af2s (and tag-free Af2) and a remarkable recovery of thermal stability ($\Delta T_m = +9$ K compared to Af2p) (Table 1, Fig. 1a–c). Mutational analysis suggests that the amino acid changes evolved during affinity maturation had a cumulative effect on the gain of affinity, with the deletion Δ78I having the largest contribution (Supplementary Discussion).

As the crystallisation of Af2 in complex with 67B89 was unsuccessful, truncated versions of the target were generated (sequences given in Supplementary Fig. S2). The binding affinities of the Fn variants 7B8 and B89 to Af2s were very similar to that of 67B89. However, 67B and the control 6789, which lacks the EDB, did not exhibit any binding (Table 1, Fig. 1c). This suggests an EDB-specific binding mode, with the domains EDB and Fn8 making the most significant contributions.

Af2s maintains binding to oncofetal fibronectin in cell culture. An immunofluorescence assay showed that Af2s binds specifically to the EDB, which is expressed at high levels by human foetal lung fibroblast Wi-38 cells. In contrast, the binding to neonatal human dermal fibroblast cells (NHDF), a cell line with low EDB expression, was significantly reduced (Fig. 1d). The absence of detectable binding of diubiquitin (variant diUb-WAA) confirms that the targeting of EDB-expressing cells is mediated by the Affilin's evolved binding properties.

### Target binding involves different register shifts in the β5 strands of Af2

The crystal structure of the Af2:7B8 complex with a resolution of 2.3 Å shows the modular architecture of the Affilin, which comprises an N-terminal Ub-like domain (Ub-N) and a C-terminal Ub-like domain (Ub-C) tethered together by a two-amino acid linker (Fig. 2a). Ub-N and Ub-C display the typical β-grasp fold of ubiquitin and bind the target 7B8 in an elongated conformation. The three Fn domains of 7B8 show the characteristic β-sandwich structure of Fn type III domains and are in slightly kinked orientation relative to each other. The structure confirms HPLC-SEC analysis data showing a complex with an apparent molecular mass of about 45 kDa, corresponding to a 1:1 stoichiometry of binding (Supplementary Fig. S4). The asymmetric unit of the crystal structure contains three Af2:7B8 complexes related by a $3_1$ non-crystallographic symmetry-axis (Supplementary Fig. S5). The superposition of the three complexes reveal a high overall structural similarity, with an average $C_\alpha$ RMSD of 3.7 Å. The Af2 molecules (chains L, J, M) show less structural differences with an average $C_\alpha$ RMSD of 2.0 Å, compared to the 7B8 molecules (chains A, B, C) with an average $C_\alpha$ RMSD of 4.3 Å. The largest deviations result from the shifted position of Fn7 in the C:M complex, as indicated by an average $C_\alpha$ RMSD of 7.0 Å to Fn7 in the A:L and B:J complexes. The three complexes demonstrate that Af2 interacts solely with the target domains EDB and Fn8, with no direct involvement of Fn7 in binding. Additional deviations are observed in the β1β2 loop and the linker region of the C:M complex. However, these differences originate from packing effects between the individual complexes in the asymmetric unit and do not imply alternative binding modes. The structural descriptions below refer to the A:L complex.

The two Ub-like domains of Af2 form a total binding interface of 1179 Å², with the major contribution from Ub-C (811 Å²) compared to Ub-N (368 Å²). Target recognition by Af2 is based on three sub-interfaces (I-III, Fig. 2a). Interface I between Ub-N and EDB includes evolved residues from the α2β5 loop and strand β1, but also native (non-evolved) amino acids from

**Table 1 | Binding characteristics and thermal stabilities of Af2 variants**

| Affilin [a,d] | DSF | ELISA | SPR | | |
|---|---|---|---|---|---|
| | $T_m$ [°C] | $K_D$ [nM] | $K_D$ [nM] | $k_{on}$ [1/M·s] | $k_{off}$ [1/s] |
| Af2p | 53 | 7.7 | 17.1 | $5.7 \cdot 10^5$ | $9.8 \cdot 10^{-3}$ |
| Af2s | 60 | 3.1 | 3.7 | $3.0 \cdot 10^5$ | $1.1 \cdot 10^{-3}$ |
| Af2 | 62 | 3.6 | 1.5 | $1.4 \cdot 10^6$ | $2.2 \cdot 10^{-3}$ |
| Af2s-Y141A | 62 | – | 83.5 | $1.1 \cdot 10^6$ | $8.8 \cdot 10^{-2}$ |
| Af2s-ΔDP-KS | 69 | – | 0.7 | $1.9 \cdot 10^6$ | $1.4 \cdot 10^{-3}$ |
| Af2-IL-2 | – | 1.5 | 2.1 | $1.1 \cdot 10^6$ | $2.3 \cdot 10^{-3}$ |
| **Target [b,d]** | | | | | |
| 67B89 | – | 3.6 | – | – | – |
| 7B8 | – | 2.6 | – | – | – |
| 67B | – | n.b.[c] | – | – | – |
| B89 | – | 2.1 | – | – | – |
| 6789 | – | n.b.[c] | – | – | – |

[a]Binding to target 67B89.
[b]Binding to Affilin Af2.
[c]No binding detected.
[d]– indicates 'not measured'.

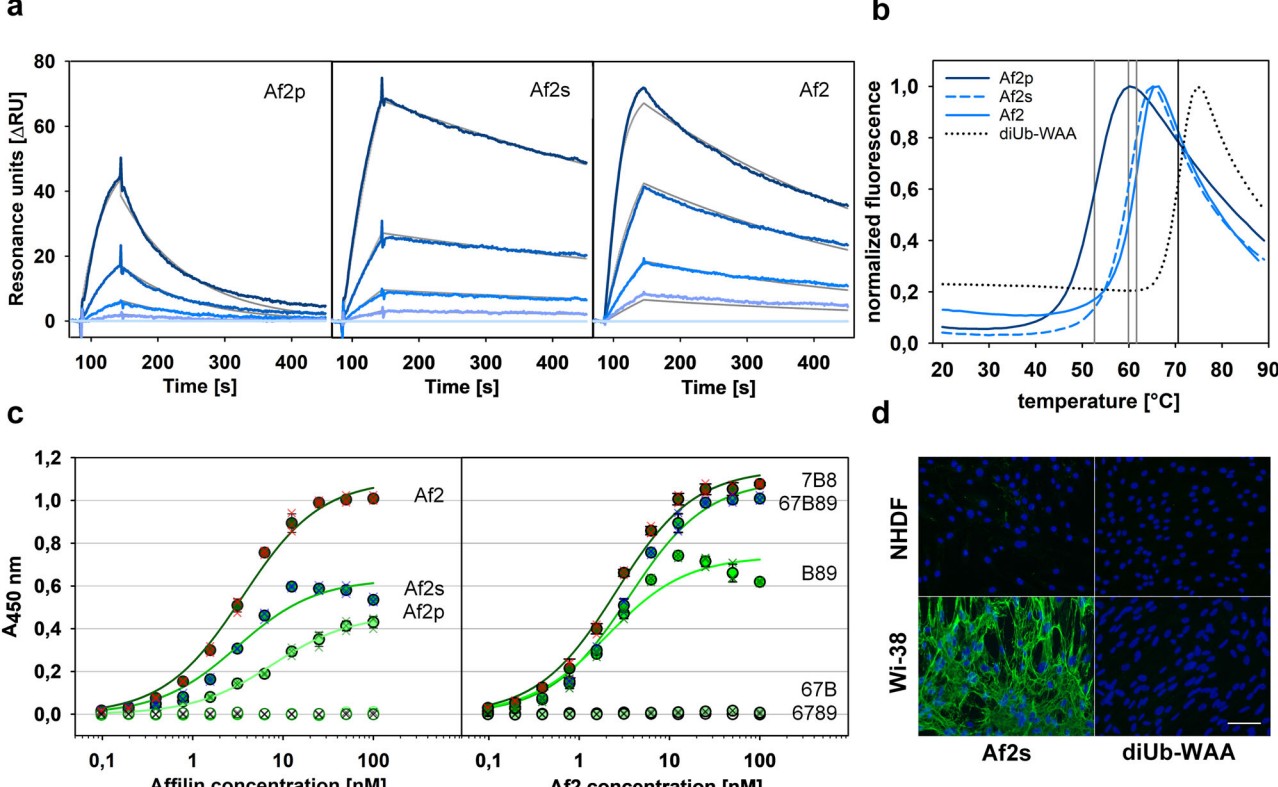

**Fig. 1 | Analysis of binding properties and thermal stability of evolved Affilin variants. a** SPR analysis of increasing Affilin concentrations binding to the immobilized target 67B89. Affilin concentrations used (sensorgram colours from light to dark blue): 0 nM, 1.2 nM, 3.7 nM, 11.1 nM and 33.3 nM. Resulting data are given in Table 1. **b** Analysis of Affilin thermal stabilities by DSF in comparison with the non-evolved parental diubiquitin variant (diUb-WAA, $T_m = 71\ °C$). Vertical lines indicate the midpoint transition temperatures ($T_m$). **c** EDB-specific binding analysed by ELISA, left panel: binding of Affilin variants to 67B89 (filled circles) and absence of binding to 6789 (open circles). Right panel: comparison of Af2 binding to different oncofetal Fn fragments (67B89, 7B8, B89), no affinity to 67B and 6789. Mean data points and error bars shown with black contours, individual measurements superimposed as coloured crosses. **d** Binding of Af2s to human foetal lung fibroblasts cells (Wi-38, high EBD expression, detected by immunofluorescence in green, cell nuclei stained in blue) and no binding of Af2s to neonatal human dermal fibroblasts (NHDF, no EDB expression). No binding is observed for (non-evolved) diubiquitin variant diUb-WAA. Scale bar: 100 μm.

β1 and β5 of Af2, which interact with EDB residues located in β-strands C/C' and loop regions BC/C'E (Fig. 2b). Interface II is formed between Ub-C and EDB, including evolved residues from the α2β5 loop and the non-evolved R149 from β5 of Ub-C that interacts with EDB residues from β-strand C' and the EF loop (Fig. 2c). Interface III involves interactions between Fn8 and Ub-C and does not comprise any evolved residues. Instead native residues, located in strands β3/β5 and the β3β4-loop of Ub-C, interact with amino acids of the FG loop of Fn8 (Fig. 2d, Supplementary Figs. S6 and S7).

The most striking observation was made by the discovery that the β5-strands of both Ub domains of target-bound Af2 have slipped from their original position in the β-sheet (as in the structure of Ub-wt[18]) by several residues towards the N-terminus (Fig. 3). The resulting negative register shifts of β5 were clearly visible in the $2F_o$-$F_c$ electron density of the Af2:7B8 complex, (Supplementary Fig. S8a, b) and are present in all complexes of the asymmetric unit. They are remarkable events due to their dramatic structural impact on neighbouring residues. In Af2, they recruit two Ub-N residues (-2 shift of L65/K66) and four Ub-C residues (-4 shift of R142/R65*, F143/F66*, L144/L67*, H145/H68*, the asterisk designating the residue numbering corresponding to Ub-N), respectively, from β5 to the preceding α2β5 loop. As these residues are directly involved in or located close to interfaces I and II, the register shifts remodel the complete target binding site (Fig. 2b, c). Interestingly, only non-evolved Af2 residues successive to β5 contribute to binding interface III. These amino acids have been relocated by the register shifts into new positions where they contribute to target binding (Fig. 2d).

The evolved residue Y141 (Y64*), located in the α2β5 loop of Ub-C, seems to have a crucial role in stabilising the two Ub domains in an orientation predestined for target binding. The sidechain of Y141 is packed in the 308 Å² interface between Ub-N and Ub-C, where it mediates domain contacts and locks the α2β5 loop in its binding conformation (Fig. 3a). Substitution of Y141 with alanine (variant Af2s-Y141A) results in a stable Affilin variant that exhibits a significantly reduced binding affinity towards the target 67B89, compared to Af2s (Table 1, Supplementary Figs. S9 and S10).

The β-strand slippages of Af2 appear to be facilitated by the alternate spacing of a series of four leucine residues (L67, L69, L71, L73) in β5, which are also present in Ub-wt. This sequence motif allows for the occupation of two hydrophobic cavities inside the Ub domain core by leucine residues, even in the −2/−4 register-shift state (Supplementary Figs. S1 and S8c, d). Thermal unfolding analysis of Af2 variants (Table 1, Fig. 1b) suggests the register shifts are associated with significant thermodynamic destabilisation compared to diubiquitin ($T_m = 71\ °C$). Amino acid deletions in the α2β5 loop, whether acquired through evolution (such as the deletion of Q140 during PD) or through rational design (variant Af2s-ΔDP-KS), seem to compensate for the loop extension induced by β5 slippage. This is exemplified by the variant Af2s-ΔDP-KS, which exhibits higher thermal stability and increased target affinity (Table 1, Supplementary Figs. S9 and S10). The rationale behind this variant and detailed description of the molecular interactions stabilising the register shifts of Af2 and the binding interface are given in the Supplementary Discussion.

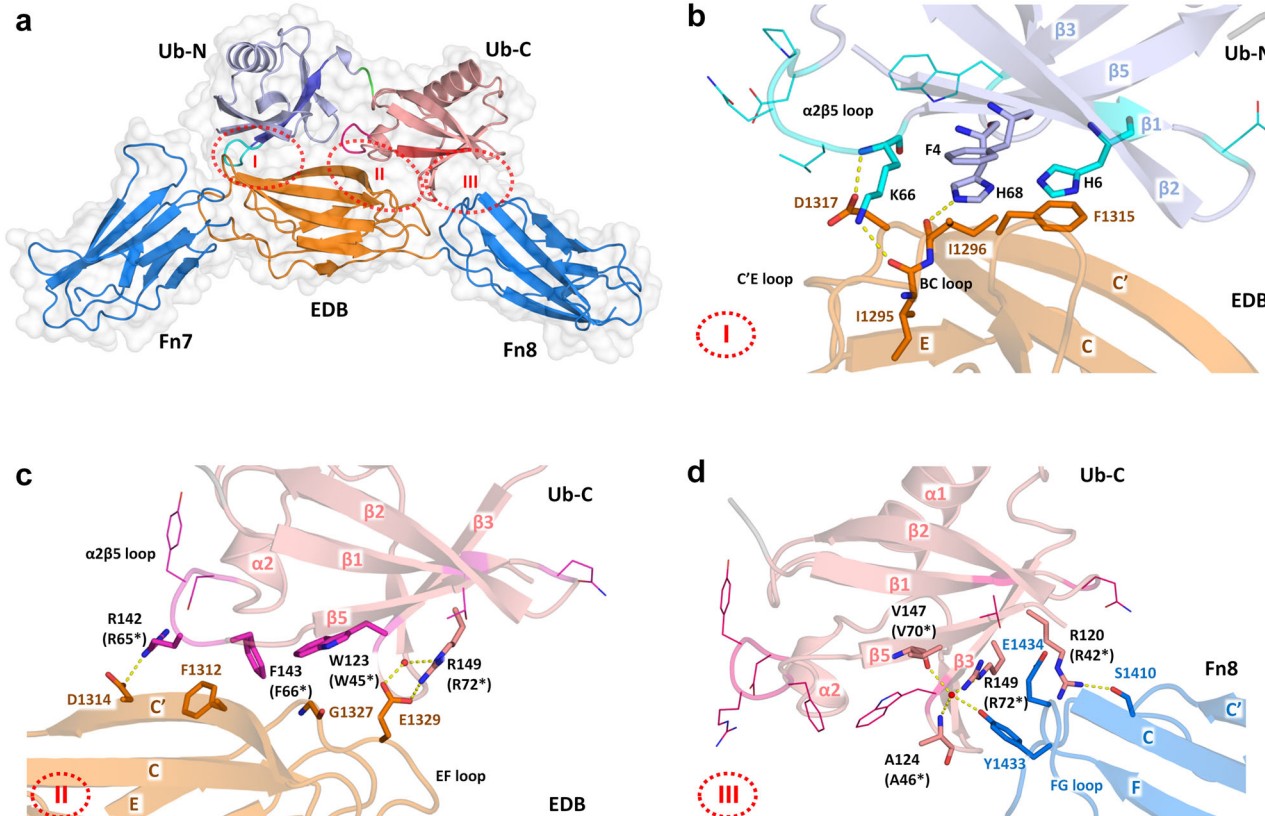

**Fig. 2 | Crystal structure of Affilin Af2 in complex with the target 7B8. a** The crystal structure of Affilin Af2 bound to oncofetal Fn fragment 7B8 revealing three target-binding regions (I-III, dotted red elipsoids) involving the Fn domains EDB and Fn8. The β5-strands of Af2 Ubi-N (darker blue) and Ubi-C (darker red) exhibit distinct negative register shifts, remodelling the α2β5 loops involved in binding (coloured in cyan and magenta, respectively). Detailed view of key interactions between (**b**) Ub-N and EDB (interface I), (**c**) Ub-C and EDB (interface II) and (**d**) Ub-C and Fn8 (interface III). Residues involved in binding shown as sticks, other evolved amino acids as lines, hydrogen bonds depicted as yellow dashed lines, water molecules as red spheres. Residue colouring of C-atoms in cyan/light blue (Ub-N), magenta/light red (Ub-C), orange/blue (EDB). Ub-C residue numbers labelled with asterisk correspond to the residue numbering of the Ub-N domain.

## The structure of Af1 reveals β5 register-shifts in absence of the target

Af1 and Af2 were independently evolved against the same target 67B89. Four of the substituted Af1 residues are identical in Af2, including H6, P63, L65 and Q86. Both Affilin variants differ at 14 positions, including three unique positions that were randomised only in Af1 (R2, W4) and Af2 (D8), respectively. The deletion at position 140 of Af2 (ΔQ140) was not observed in Af1 and two changes were uniquely evolved in Af2 during the affinity maturation (Q38, ΔI78 in the linker, full sequences of Af1 and Af2 given in Supplementary Fig. S1). Attempts to crystallise the purified Af1:7B8 complex resulted in crystals of Af1 alone that diffracted to 2.2 Å, allowing elucidation of the Af1 structure in the unbound state (Fig. 4a). The superposition of Af1 with the Af2:7B8 complex shows that the two Ub domains have different orientations. The Ub-C of Af1 is rotated approximately 180° in relation to Ub-N, around an axis almost parallel to the three-amino acid linker (Fig. 4b). Solvent-exposed W142 of Af1 is not involved in domain contacts between Ub-N and Ub-C, as observed for the corresponding Y141 in Af2.

Both Af1 domains exhibit a -2 register shift in β5, proving that the strand slippage is present even in the absence of the target. Af1 Ub-N und Ub-C display almost identical overall structures (Cα RMSD 0.2 Å, Fig. 4c). The molecular interactions stabilising the −2 register shifts of Af1 are similar to those in Af2 Ub-N (described in detail in the Supplementary Discussion). The very similar structures of the Ub-N domains of Af1 and Af2 (Cα RMSD 1.5 Å) both display a −2 register shift in β5. However, there are differences in the conformations of the α2β5 loops (Fig. 4d). The

structural superposition of the Ub-C domains of Af1 and Af2 (Cα RMSD 2.5 Å) illustrates the plasticity of the Affilin scaffold. The individual register shifts (−2/−4) remodel the α2β5 loop differently, depending on the number of accommodated residues and results in corresponding retractions of the C-terminal residues towards the target binding site (Fig. 4e).

Currently, the structural basis of target binding by Af1 remains elusive due to the unavailability of a complex structure. Our attempts to predict the β5 register shifts of Af1 and Af2 using Alphafold2[19] have been unsuccessful, both in the absence and presence of the target 7B8. In none of the predicted structures register shifts of β5 were observed, precluding the prediction of a potential Af1 complex structure (Supplementary Fig. S13).

Presumably, the −2 register shift of Af1 Ub-C is incompatible with the binding mode observed in the Af2:7B8 complex, which involves a −4 shift in Ub-C, and would result in clashes of Af1 Ub-C with the target. In contrast, the Ub-N domains of Af1 and Af2 share the same register shift and could allow similar interactions to the EDB (Supplementary Fig. S14). Three different binding modes for Af1 are conceivable: (i) Af1 binds the target completely differently to Af2; (ii) Af1 Ub-N interacts with the target as in the Af2:7B8 complex, while Af1 Ub-C recognizes a different epitope of oncofetal fibronectin; or (iii) Ub-N and Ub-C of Af1 bind the target similarly to the Af2:7B8 complex, which would require additional conformational changes in Ub-C. In the last scenario, Af1 Ub-C would need to undergo a significant domain reorientation to contact the target domains. Additionally, a binding-induced extension of the register shift from −2 to −4 would be required, recruiting Af1 residues into positions appropriate for target binding.

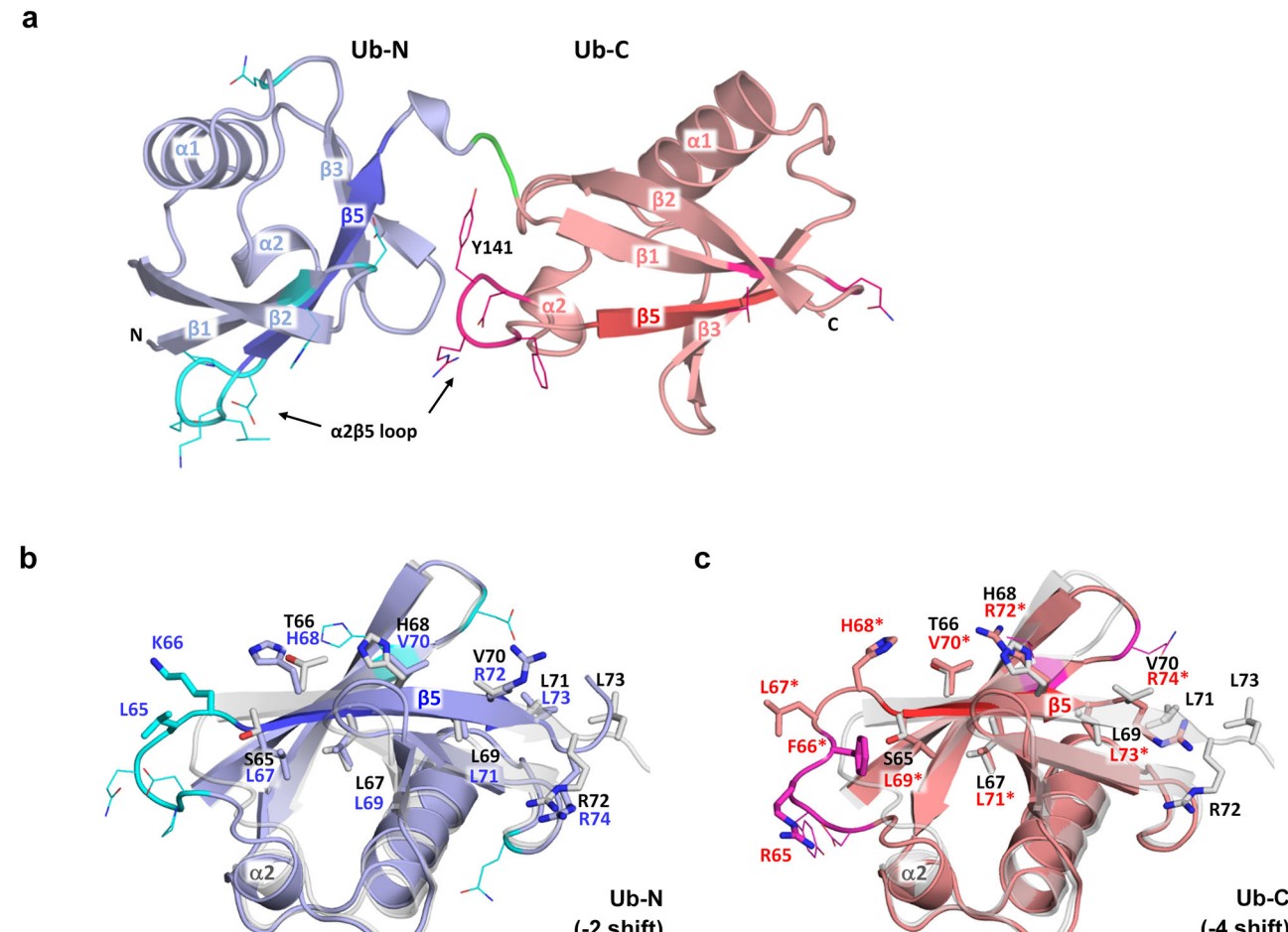

**Fig. 3 | The different β5 register shifts observed in the 7B8-bound structure of Af2. a** Overall structure of 7B8-bound Af2 depicting the distribution of the evolved residues of Ub-N and Ub-C (side chains shown as lines, C-atoms of Ub-N cyan, C-atoms of Ub-C magenta). Linker residues coloured in green. The β5 strands of Af2 Ubi-N (darker blue) and Ubi-C (darker red) exhibit distinct negative register shifts, extending the α2β5 loops involved in target binding. Structural superposition of Ub-wt (PDB id 1UBQ, light grey, black labels) and (**b**) the Ub-N domain of Af2 (light blue, blue labels, evolved residues shown in cyan) revealing the -2 register-shift of β5 (darker blue) and (**c**) the Ub-C domain of Af2 (light red, red labels with asterisk, corresponding to residue numbering of Ub-N domain, evolved residues shown in magenta) exhibiting a -4 register-shift of β5 (darker red). Af2 residues relocated by the β5 register shifts and corresponding residues of Ub-wt shown as sticks.

## Generation of a functional Af2-Interleukin-2 chimera targeting oncofetal fibronectin

Delivery of the cytokine IL-2, to sites of tumoral angiogenesis, can augment cellular cytotoxicity by local T-cell stimulation[20,21]. In conceptual analogy to immunecytokines, human IL-2 was genetically fused as a payload to Af2 (Af2-IL-2). As the expression of the fusion protein in *E.coli* resulted in the formation of insoluble inclusion bodies, in vitro refolding was employed to facilitate protein production, followed by standard purification procedures. The functionality of Af2-IL-2 was assessed by analysing the binding to the target 67B89, revealing affinity in the low nanomolar range with binding kinetics very similar to Af2. The specificity towards the target 67B89 was preserved, as no binding to 6789 could be observed (Table 1, Fig. 5a, Supplementary Fig. S9e). The Af2-IL-2 fusion can bind specifically to Wi-38 cells (a cell culture of human foetal lung fibroblasts expressing a high level of oncofetal fibronectin) as shown by an immunofluorescence assay (Fig. 5b). In contrast, binding to a cell line with low EDB expression (NHDF) was negligible. Cytokine activity was assayed using a cell line of IL-2-dependent murine cytotoxic T-cells (CTLL-2) with recombinant human IL-2 as a reference. The comparison of mean $EC_{50}$ values showed similar potencies of refolded Af2-IL-2 ($EC_{50} = 51 \pm 4$ pM) and recombinant IL-2 ($EC_{50} = 46 \pm 5$ pM) (Fig. 5c). This demonstrates that the chimeric Affilin-cytokine fusion is capable of

targeting EDB-expressing cells in vitro and that the activity of the payload IL-2 is not significantly affected in context of Af2.

## Discussion

The directed evolution of proteins with engineered binding properties has employed many non-antibody scaffolds successfully, circumnavigating the structural complexity of immunoglobulins and derived molecules[11]. For the most part, these artificial binding proteins have adopted the natural paradigm of antibody diversity, where structural adaptations to target binding are achieved predominantly via variable (evolved) residues framed in a rigid scaffold. Confirmed in numerous structural studies, this principle governs scaffold selection as well as library design[13]. With the crystal structures of the two independently evolved Affilin variants Af1 and Af2, raised against the target oncofetal fibronectin, we have determined the structural basis of target recognition by a diubiquitin-based artificial binding protein, revealing a striking example of scaffold plasticity (Figs. 2–4). We observed variations of the β-sheet register that result from apparent β5 strand slippages by 2 or 4 residues. Negative register shifts (towards the N-terminus) were found for each Ub domain of unbound Af1 (-2 shifts in Ub-N and Ub-C) and target-bound Af2 (-2 shift in Ub-N, -4 shift in Ub-C), respectively. The structural consequences of the shifts are two-fold. Firstly, the α2β5 loop, which is in direct contact with the target 7B8, is extended by two (-2 shift) or three

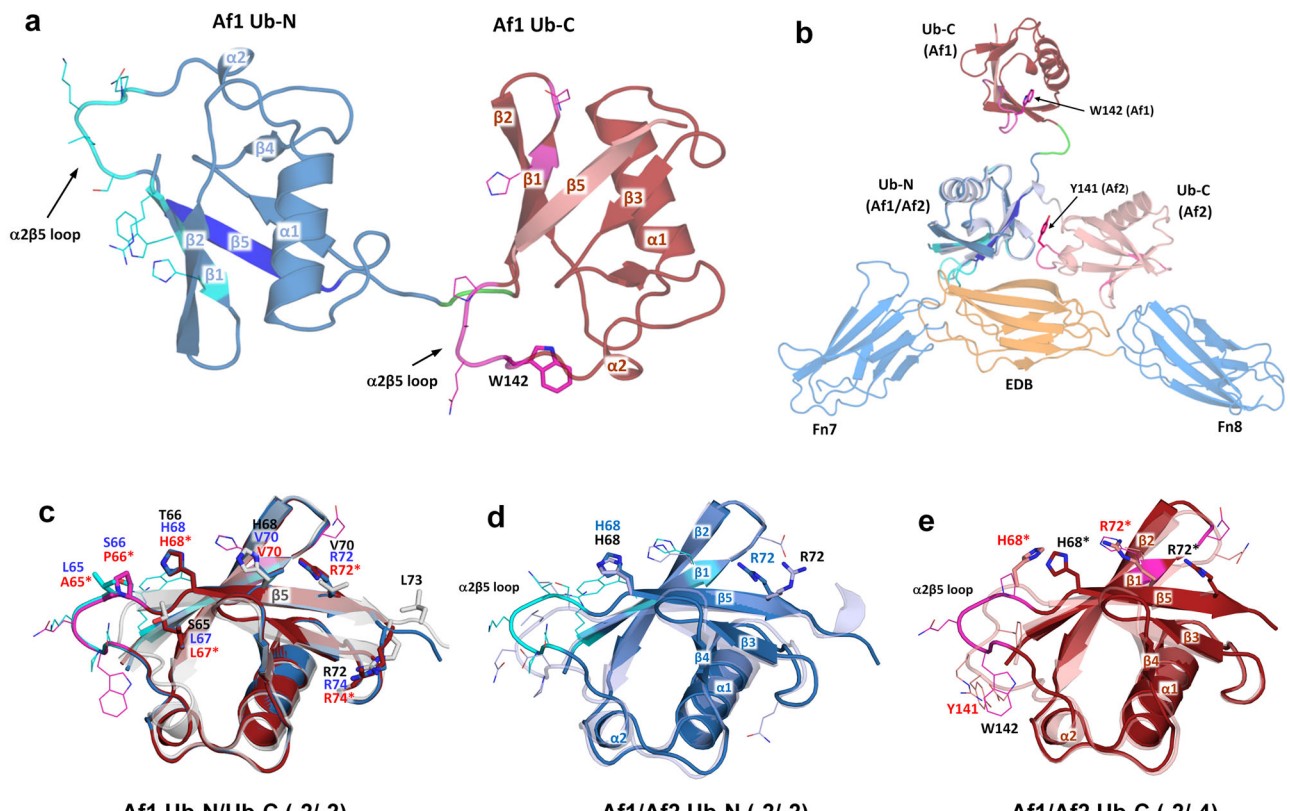

**Fig. 4 | Crystal structure of unbound Affilin Af1. a** Overall structure of Af1 depicting the distribution of the evolved residues of Ub-N and Ub-C (side chains shown as lines, C-atoms of Ub-N cyan, C-atoms of Ub-C magenta). Linker residues coloured in green. The β5 strands of Af1 Ub-N (bright blue) and Ub-C (lighter red) exhibit -2 negative register shifts. W142 (corresponding to Y141 in Af2) shown as sticks. **b** Structural superposition of Af1 and the Af2:7B8 complex, based on the alignment of the Ub-N domains, showing the different relative orientations of Ub-N und Ub-C. Regions of evolved residues coloured in cyan (Ub-N) and magenta (Ub-C). Y141 (Af2) and W142 (Af1) shown as sticks. **c** The two β5 register shifts of Af1 are apparent from the structural superposition of Ub-wt (PDB id 1UBQ, light grey, black labels) with Af1 Ub-N (blue, blue labels) and Ub-C (dark red, red labels). Residues of β5 shown as sticks, evolved residues outside β5 shown as lines. Structural comparison of (**d**) the Ub-N domains of Af1 (dark blue, black labels) and Af2 (light blue, blue labels), both exhibiting -2 register shifts and (**e**) the Ub-C domains of af Af1 (dark red, black labels) and Af2 (light red, red labels) with distinct β5 register shifts (−2/−4). Ub-C residue numbers labelled with asterisk correspond to the residue numbering of the Ub-N domain. Residues H68/H68* and R72/R72*, shown as sticks, mark the different structural rearrangements around β5 caused by the register shifts.

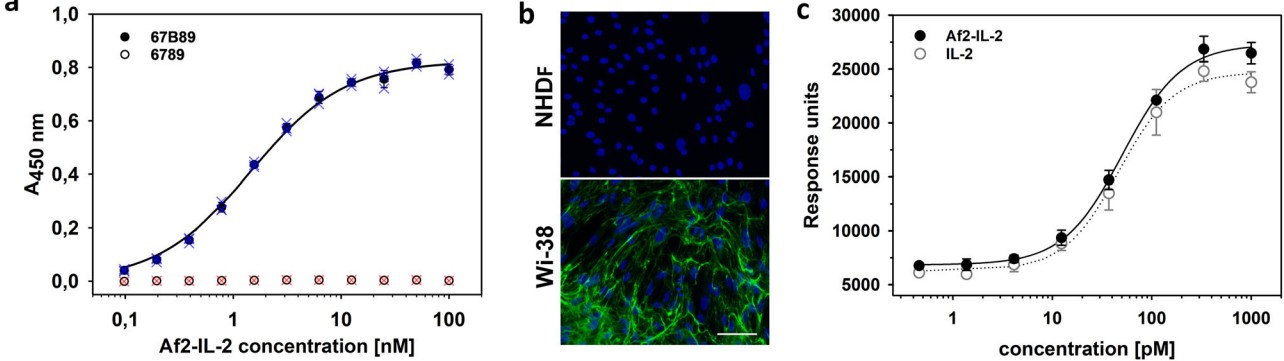

**Fig. 5 | Binding and functional analysis of an EDB-specific Affilin-IL-2 fusion. a** Binding of Af2-IL-2 to the target 67B89 (filled circles) and lack of binding to the off-target 6789 (open circles) analysed by ELISA (binding parameters given in Table 1). Mean data points and error bars shown in black, individual measurements superimposed as coloured crosses. **b** Binding of Af2-IL-2 to human foetal lung fibroblasts cells (Wi-38, high EBD expression, detected by immunofluorescence in green, cell nuclei stained in blue) and no binding of Af2-IL-2 to neonatal human dermal fibroblasts (NHDF, no EDB expression), scale bar: 100 μm. **c** IL-2-dependent growth of murine cytotoxic T-cells (CTLL-2) after application of Af2-IL-2 (filled black circles, black error bars), recombinant human IL-2 (open grey circles, grey error bars) serving as reference.

residues (-4 shift, coinciding with the evolved deletion ΔQ140 in Af2), which were previously located in β5. Secondly, all subsequent domain residues are relocated, recruiting non-evolved residues to positions where they can contribute to target binding.

The binding mode elucidated from the Af2:7B8 complex structure can rationalise the observations from the binding analysis (Figs. 1 and 2). Af2 binds to the Fn variants 7B8, 67B89 and B89 with high affinity through binding interfaces I to III that involve solely the EDB and Fn8. In contrast, Af2 does not exhibit significant affinity to 67B, despite the theoretical possibility of binding the EDB through interfaces I and II. However, in addition to EDB, Fn8 is also crucial for the interaction, indicating that the engagement of interface III is necessary for either the correct alignment of the Ub-N and Ub-C domains or the structural stabilisation of the -4 register shift in target-bound Ub-C, or both. Likewise, the absence of the EDB and therefore interfaces I and II, which provide the major molecular surface area (853 Å$^2$) for binding, can explain the lack of Af2 binding to 6789. Although Fn8 could theoretically engage interface III (326 Å$^2$), it is conceivable that in the absence of the EDB, a -4 register shift (stabilised by target binding) may not be present, rendering the domain Ub-C binding-incompetent.

Structural evidence for β-strand slippage in proteins is sparse. These backbone rearrangements are not easily predictable from the sequence and may even be missed in (lower resolution) experimental structures. Recent structure prediction algorithms, including Alphafold2, are currently unable to predict the β-strand register shifts observed in the Affilin crystal structures, as our attempts have shown. However, these events could be more common than current structural knowledge reflects. Wild-type ubiquitin is a prototype example, where it has been shown that β5 strand slippage is a natural feature of ubiquitin[22]. During mitophagy (the clearance of damaged mitochondria), Ub-wt is phosphorylated at S65 by the Ub kinase PINK1[23]. This requires a transient -2 register shift of the Ub β5 strand to relocate S65 to an exposed position in the α2β5 loop, allowing PINK1 access to the substrate residue. In the absence of phosphorylation, the population of the -2 register-shifted state of Ub-wt is low (<1%), whereas phosphorylation by PINK1 or mutations of Ub residues in β5 (e.g. T66V and L67N, variant Ub-TVLN) shift the conformational equilibrium significantly towards the register-shifted state[22,24–26] (Supplementary Fig. S15a). A register shift of more than two residues, as found in Ub-C of the Affilin Af2, has been reported only once before for an evolved ubiquitin variant (Ubv-G08) possessing 7 mutations spread over the strands β1, β4, β5 and the α2β5 loop. The crystal structure of Ubv-G08 in complex with its target revealed a −4 register shift of the β5 strand, with concomitant extension of the α2β5 loop, with both regions extensively involved in ligand binding[27] (Supplementary Fig. S15b).

Slippage of β-strands has been described for a number of other proteins. GTP-binding proteins link strand slippage to GTP hydrolysis affecting membrane remodelling and filament formation[28,29]. The structural changes induced by a three-residue negative register shift of a β-strand in human transthyretin, a tetrameric plasma protein associated with amyloidosis, are directly linked to amyloid fibril formation[30]. A similar three-residue register shift was observed during the activation of the zymogen coagulation factor VII[31]. Bacterial flavin-dependent BLUF photoreceptors employ β-strand slippage in the transition between dark and light state[32] and in the *Vibrio cholerae* toxin HigB2 a single-residue β-strand register shift shuts off its mRNAse activity by flipping a catalytically important residue out of the active site[33].

β-strand register shifts represent one facet of a more general plasticity of β-sheets, although this may not surface for every fold containing β-sheets. However, well-studied examples are the rearrangement of β-sheets in donor strand complementation, both transient and permanent, reported in processes such as bacterial pilus formation[34], protease inhibition by serpins[35], ubiquitin ligase assembly[36] and chaperone activity[37]. Another example are domain-swapped proteins that are often formed through wholesale exchange of β-strands between molecules[38]. The associated structural events involve reorganization of hydrogen bonding and side chain packing similar to that required for β-strand slippage.

Register shifts of β-strands may provide a conceptual framework for the evolution of sequence insertions and deletions (InDels) at the protein structure level to generate stable neofunctional domains. They add a fascinating aspect to (directed) protein evolution beyond localized mutations, as these events place evolved residues into new positions and recruit neighbouring amino acids into new environments. However, designing InDels rationally remains a complex task, despite significant progress in implementing them into protein engineering approaches[12]. Protein backbone plasticity that allows for the generation of InDels, for instance through β-strand slippage, might help simplifying this task. These events can enhance the structural diversity of the candidate pool without the need to expand the size of the genetic library. Although speculative at this point, in the case of the Affilin design, the two tethered Ub domains could potentially exhibit three different register-shifted states (0, −2, −4) each, resulting in a ninefold increase in the number of structural variants in total.

The structure determination of the oncofetal-fibronectin-specific Affilin variants Af1 and Af2 presented here, demonstrate that scaffold plasticity was crucial in obtaining high affinity binders, as β-strand register shifts have been observed in all four Ub domains. They have co-evolved with amino acid composition under the selection pressure of directed evolution. The structures also exemplify potential caveats in the interpretation of results from combinatorial protein engineering and underline the importance of structural analyses. As demonstrated for the Affilin molecules, such endeavours can uncover unforeseeable structural alterations. Our data add to previous reports on artificially evolved binding proteins, which describe for instance unexpected binding modes for protein Z-based affibodies[39] and a single-domain antibody fragment[40]. However, they also demonstrate that the intrinsic plasticity of these proteins can indeed be exploited – and possibly extended - by directed evolution. Finally, combining the *de-novo* binding properties of an evolved Affilin with the function of a payload protein, as demonstrated for the Affilin-IL2 chimera, offers new perspectives towards their use in medical and biotechnological applications.

## Methods
### Production of fibronectin fragments
The genes for the target proteins, human fibronectin fragments containing EDB (67B89, 67B, 7B8 and B89, Uniprot ID P02751, isoform 7, variant C1232S) and off-target fragment 6789, lacking the EDB, were obtained via gene synthesis (Geneart, Regensburg, Germany) and cloned into pET28a expression vector. The plasmids were transferred into *E. coli* BL21 (Lucigen, Middleton, WI, USA). Fn fragments 67B89, 6789 and 7B8 were expressed without any affinity tags, 67B contained an N-terminal His-tag and B89 a C-terminal His-tag, respectively (sequences given in Supplementary Fig. S2). After protein expression the untagged variants (67B89, 6789 m 7B8) were purified by anionic exchange (Q-Sepharose FF XK 26/20 column), ammonium sulfate precipitation, hydrophobic interaction chromatography (HiTrap Phenyl HP column) and size exclusion chromatography (Superdex 200 XK 26/60 column). The His-tagged fragments 67B and B89 were purified by immobilized metal affinity chromatography (IMAC, HisTrap HP, GE Healthcare), according to the manufacturers recommendations, followed by size exclusion chromatography (Superdex 200 XK 26/60 column). All chromatographic steps were carried out on an Aekta Explorer system (GE Healthcare, Freiburg, Germany).

To allow selection of binders by ribosome and phage display, preferential N-terminal biotin labelling of the target 67B89 was achieved after dialysis of the sample against 50 mM sodium phosphate buffer pH 6.5 and subsequent incubation with a 30-fold molar excess of EZ-Link Sulfo-NHS-LC-Biotin reagent (Pierce, Rockford, IL, USA) for 24 h at 4 °C. Non-coupled reagent was removed by dialysis against PBS pH 7.4.

### Affilin library construction
The Affilin scaffold used in this work consists of a linear fusion of two ubiquitin molecules (Uniprot ID P0CG47, residues 1–76). Prior to library generation three point mutations were introduced to improve

spectrophotometric sensitivity[41] and manufacturability (F45W, G75A, G76A, yielding ubiquitin variant Ub-WAA). In silico analysis of protein stability effects exerted by mutation of candidate surface-exposed residues of ubiquitin identified 9 amino acid positions (2, 4, 6, 8, 62–66) which were selected for randomisation in Affilin Af1, as described previously[15]. For Affilin Af2, two library modules of monomeric Ub-WAA incorporating random amino acids (except cysteine) at positions 6, 8 and 62–66 were synthesized, differing only in codon usage (Morphosys, Martinsried, Germany). Introduction of *Mfe*I and *Eco*RI restriction sites via PCR, followed by digestion and ligation of the amplified fragments yielded a library of diubiquitin (diUb-WAA), resulting from the linear fusion of both Ub-WAA monomer libraries separated by a Gly-Ile-Gly linker. The diubiquitin library was amplified via PCR to insert *Bsa*I restriction sites. Following digestion with *Bsa*I the insert was purified using magnetic streptavidin coupled beads (M-270 Dynabeads, Life Technologies, Carlsbad, CA) and subsequently ligated to phagemid pCD12, a derivative of phagemid pCD87SA[42]. *E. coli* ER2738 cells (Lucigen, Middleton, WI) were transformed with the resulting phagemids by electroporation followed by single-colony PCR and DNA sequencing to assess correct size and sequence of the inserts. All transformed clones were purified using the QIAfilter Plasmid Maxi Kit (Qiagen, Hilden, Germany) to obtain the phagemid library. A library size of $2.5 \cdot 10^9$ variants was calculated as effective, taking into account the constraining effects of frame-shifts and limited transformation efficiency.

### Phage display selection and screening of EDB-specific Affilin variants

Generation of Af1 is described elsewhere[15,43]. For Af2 precursors, Tat-mediated phage display (PD)[42] selection and screening of was performed at 20 °C in a similar manner. Briefly, N-terminally biotinylated 67B89 target protein was immobilized on Streptavidin Dynabeads M-270 (Invitrogen, Carlsbad, CA, USA). The target-coated beads were blocked with BSA and incubated with a suspension of $3.4 \cdot 10^{12}$ phages in the presence of a 10-fold molar excess of the off-target (variant 6789), followed by washing the beads with PBST. To further increase selection pressure, the amount of immobilized target protein was decreased within two subsequent PD iterations while washing stringency was increased during the four panning rounds. Bound phages were cleaved by addition of 30 µg/mL trypsin (Roche Diagnostics, Mannheim, Germany). Phagemids of the selected binding molecules were cloned into pPR-IBAF1b vector (IBA, Goettingen, Germany) to yield expression constructs of the binders with C-terminal Strep-tag II. Following transformation of *E. coli* BL21 (DE3) single colonies were picked for screening of candidate binders and cultivated in 96-well scale. Cells were harvested by centrifugation and pellets were suspended in PBST with 250 µg/mL lysozyme (Merck, Darmstadt, Germany) and lysed by three subsequent freeze-thaw-cycles. To select for stable binders the resulting lysates were incubated at 50 °C for two hours leading to heat precipitation of thermodynamically instable variants. After centrifugation screening of soluble EDB-specific binders was performed by ELISA as follows: lysates were incubated with target-coated (67B89) and off-target-coated (6789) 96-well Medisorp-plates (Nunc, Roskilde, Denmark) followed by washing with PBST and PBS. Bound Affilin molecules were detected using an anti-Ubi-Fab-HRP conjugate (AbD Serotec, Puchheim, Germany) using TMB Plus (Kem-En-Tec Diagnostics, Taastrup, Denmark) as substrate. Variants having a target/off-target binding ratio of >2 were defined as binders.

### Affinity maturation, ribosome display and screening of EDB-specific Affilin variants

Affinity maturation of the precursor variant Af2p, to generate the improved Affilin Af2, was conducted as a sequence of random mutagenesis followed by a selection of binders using ribosome display (RD). The cDNA of Affilin variant Af2p was used as template for error-prone PCR employing the GeneMorph II Random Mutagenesis Kit (Agilent, Santa Clara, CA, USA). The error rate was set to 10–14 mutations per kbp and the resulting theoretical library size was calculated to contain $6 \cdot 10^{10}$ variants. Linker segments comprising functional elements required for ribosome display were fused

via PCR to the 5' and 3'-ends of the generated library[44]. Subsequently, four cycles of RD were performed using the PureExpress in vitro protein synthesis kit (New England Biolabs, Ipswich, MA, USA) for in vitro transcription and translation. Ternary complexes were incubated with the biotinylated target 67B89 in PBSNT (PBS supplemented with 30 mM magnesium acetate and 0.05% v/v Tween 20) and a 10-fold molar excess of (non-biotinylated) off-target 6789. Target-bound complexes were recovered using M-270 Streptavidin beads (Invitrogen, Carlsbad, CA, USA). Stringent selection of high-affinity binders was achieved by up to 6 washing steps with PBSMT and competitive elution of the immobilized complexes using decreasing concentrations of non-biotinylated target 67B89 in the last two cycles of RD. After each cycle the mRNA was released by addition of EDTA and subsequently reverse-transcribed to obtain the corresponding cDNAs. Prior to the next cycle the cDNA of the pool of binders was re-amplified and supplied with the RD-linker segments by two consecutive PCR reactions. After the fourth cycle of RD the cDNAs of the selected binders were cloned via *Nde*I/*Xho*I restriction sites into an expression vector (pET-Strep-eGFP) providing a genetic fusion of enhanced GFP (eGFP) to the C-terminus of the Affilin variants for screening. Based on the green fluorescence intensity of single colonies of *E.coli* BL21 (DE3) expressing functional Affilin-eGFP fusions without frameshifts, candidates were selected using a K3-XL colony picker (KBiosystems, Basildon, UK) for ELISA binding analysis (as described for the screening after PD). Affilin variants having a target/off-target binding ratio of >2 were defined as binders. For expression without C-terminal eGFP the cDNA of selected Affilin variants were cloned into the expression vector pPR-IBAF1b (IBA, Goettingen, Germany) providing a C-terminal Strep-tag II.

### Production and purification of Affilin variants

The recombinant expression and purification of Af1 is described elsewhere[15]. Variants of Af2p and Af2s carrying a C-terminal Strep-tag II (sequences given in Supplementary Fig. S1a) were expressed in *E.coli* BL21 (DE3) in 1-liter scale, followed by purification using a StrepTactin Superflow column (IBA, Goettingen, Germany) according to the manufacturer's instructions. A second purification step was carried out as size exclusion chromatography on a Superdex 75 pg XK16/600 column, equilibrated in in PBS pH 7.4, using an ÄKTAexpress FPLC system (GE Healthcare, Freiburg, Germany). The tag-free Affilin variant Af2, used for binding analysis and crystallisation, was generated by cloning the cDNA into the expression vector pET-20bDoSto, a derivative of pET-20b(+) (Novagen, Darmstadt, Germany) that carries an additional stop codon, resulting in omission of the C-terminal hexahistidine tag. Plasmids were subsequently transferred into electro-competent *E. coli* BL21 (DE3) cells. Expression was carried out in 1-liter scale. After cell harvest and cell disruption, proteins were purified from lysates via a HiTrap Q Sepharose FF anion exchange column and subsequent HiTrap Phenyl HP hydrophobic interaction chromatography. Purified Affilin variants were then dialysed against PBS pH 7.4.

### Production of the Af2-Interleukin-2 fusion

EDB-specific Affilin variant Af2 was genetically fused to the N-terminus of a synthetic gene of human Interleukin-2 (UniProtKB ID P60568, residues 21-153, variant C125S) separated by a 15 amino acid $(Ser_4\text{-}Gly)_3$ linker (complete sequence given in Supplementary Fig. S1b). The constructs were cloned into the vector pET-28aS, a derivative of pET-28a (Novagen, Darmstadt, Germany), via *Bsa*I-HF restriction sites, providing a tag-free expression. Resulting plasmids were transferred into electro-competent *E. coli* BL21 (DE3) cells and protein expression carried out for 4 h in 1 L scale at 37 °C. Insoluble protein expression required in vitro refolding of Af2-IL-2. Based on established protocols[45,46], harvested cells were disrupted and inclusion bodies were isolated and solubilized in 6 M guanidine hydrochloride, 100 mM Tris pH 8.5, 1 mM EDTA, 100 mM DTT. Renaturation was carried out by rapid dilution pulses of the solubilized protein (final concentration 100 µg/mL) into 1 L buffer containing 50 mM Tris pH 9.0, 3 M urea, 2.5 mM GSH, 0.25 mM GSSG) at 4 °C under gentle stirring for 16 h. Subsequent purification of the refolded protein was achieved by

addition of $(NH_4)_2SO_4$ (1 M final concentration) followed by filtration, purification via hydrophobic interaction chromatography (HiTrap Phenyl HP column) and size exclusion chromatography (XK26/600 Superdex 75 prep grade, equilibrated in PBS pH 7.4).

### Binding analysis by Surface Plasmon Resonance

Surface plasmon resonance (SPR) measurements on a Biacore 3000 (GE Healthcare, Freiburg, Germany) were used to analyse binding of purified Affilin variants to the target 67B89. Purified biotinylated target was immobilized on a streptavidin chip (GE Healthcare) according to the manufacturer's instructions. The off-target 6789 was immobilized to the reference channel of the chip. Different concentrations of the Affilin variant (0–33.3 nM) were analysed for binding to 67B89 using PBS pH 7.4 containing 0.005% Tween 20 as running buffer at a flow rate of 30 µl/min. Traces were corrected by subtraction of the reference signal and the trace of buffer injection. $K_D$, $k_{on}$ and $k_{off}$ values were calculated by fitting the traces using a global kinetic fitting (1:1 Langmuir model, BIAevaluation software).

### Binding analysis by ELISA

Determination of binding affinity by ELISA was carried out in 96-well medium binding plates (Microlon 200, Greiner Bio-One, Kremsmuenster, Austria). Plate coating with 5 µg/ml target 67B89 (or the target variants 67B, 7B8, B89, respectively) and off-target 6789 was performed by overnight incubation at 4 °C. The wells were washed three times with PBST and blocked with 3% BSA solution for 2 h at 20 °C. Binding to the target-coated plate was performed in concentration-dependent manner by incubation with the binder at concentrations up to 100 nM followed by washing the plates three times with PBS. Bound Affilin molecules were detected using an anti-Ubi-Fab-HRP conjugate (0.65 mg/ml, AbD Serotec, cat. No. AbyD03925, Puchheim, Germany) using TMB Plus (Kem-En-Tec Diagnostics, Taastrup, Denmark) as substrate. POD activity was measured spectrophotometrically at a wavelength of 450 nm in a microplate reader (Sunrise, Tecan, Maennedorf, Switzerland). Binding to the target 67B89 was determined in triplicates and by single measurement for the off-target 6789, respectively.

### Analysis of thermal stability

Thermal unfolding transitions of proteins were measured by means of differential scanning fluorimetry (DSF), performed at a protein concentration of 0.1 mg/ml protein in PBS pH 7.4 using a 10-fold dilution of SYPRO Orange (Invitrogen, Carlsbad CA, USA) in a real-time PCR device (Light Cycler 480, Roche Diagnostics, Mannheim, Germany). Fluorescence was recorded at 465 nm excitation and 580 nm emission wavelengths, respectively, over a temperature range of 20–90 °C with 1 K/min heating rate. For evaluation, fluorescence intensity was plotted against the temperature and the inflection point ($T_m$) derived from the maximum of the first derivative of the plot as the midpoint of thermal unfolding.

### Cell binding analysis by immunofluorescence

The Affilin variants Af2s and Af2-IL-2 were analysed for binding to EDB expressing Wi-38 cells (human foetal lung fibroblasts, ATCC CCL-75) using NHDF cells (neonatal human dermal fibroblasts, Promocell, Heidelberg, Germany), lacking EDB expression, as negative control. Cells (30,000 per well) were disseminated and cultivated for 96 h in 4-well chamber slides in 90% EMEM medium supplemented with 10% FBS (Wi-38 cells) and 98% Fibroblast Growth Medium/2% supplement mix (Promocell, NHDF cells), respectively. After washing three times with PBS, fixation with methanol, washing and blocking (5% horse serum in PBS) cells were incubated with 50 nM Af2s and Strep-tagged diUb-WAA (as non-binding control), respectively, for 1 h at 37 °C. For detection of bound Af2s, a rabbit anti-Strep-tag IgG antibody (0.5 mg/ml, Genscript, cat. No. A00875, Piscataway, NJ, USA) and a goat anti-rabbit IgG Alexa 488 conjugate (2 mg/ml, Invitrogen, cat. No. A11008, Carlsbad CA, USA) as secondary antibody were used. Detection of bound Af2-IL-2 was performed using a rat anti-human IL-2 mAb Alexa Fluor488 conjugate (Invitrogen, Carlsbad CA,

USA). Cell nuclei were counterstained with DAPI (Sigma Aldrich, Steinheim, Germany) and embedded in the polyvinyl alcohol Mowiol. Visualization was conducted using an Axio Scope AF1 fluorescence microscope (Zeiss, Jena, Germany) employing EX BP 470/40, BS FT 495 and EM BP 525/50 filters. For DAPI visualization filters EXG365, BS FT 395 and EM BP 445/50 were used.

### Activity assay of Interleukin-2 fusions

The IL-2 activity assayed using murine cytotoxic T-cells (CTLL-2, ATCC TIB-214) dependent on IL-2 for growth[47]. Cultivation was performed in full medium (78% RPMI1640, 10% FBS, 10% T-STIM, 2 mM L-glutamine, 1 mM sodium pyruvate). Harvested cells were washed twice with medium without T-STIM. Subsequently, 40,000 cells were seeded per cavity of a 96-well plate and incubated for 20 h with serial dilutions (1000 – 0.076 pM) of the fusion protein Af2-IL-2 and the reference recombinant human IL-2 (PeproTech, Rocky Hill NJ, USA), respectively. Viable cells were detected using WST-1 reagent (Roche Diagnostics, Mannheim, Germany) by absorption measurement at a wavelength of 450 nm with a reference wavelength of 620 nm. $EC_{50}$ values from triplicate measurements were calculated using the program SigmaPlot (Sysstat Software, Palo Alto, USA).

### Protein crystallization and structure determination

Complex formation of tag-free Affilin variants Af1 and Af2, respectively, with the truncated target 7B8 was accomplished by incubation of equimolar concentrations of the binding partners at 20 °C for 1 h. The complex was isolated from unbound species by size exclusion chromatography (Superdex 75 26/600) in 10 mM HEPES, 100 mM NaCl pH 7.3 and concentrated to 22 mg/mL using Amicon Ultra-4 centrifugal filters (Millipore, Billerica MA, USA).

Unbound Af1 was crystallised during attempts to obtain complex crystals of Af1 with the target 7B8. Small bipyramidal crystals were obtained within 2 weeks from hanging drop vapour diffusion crystallisation setups of the Af1:7B8 complex at 15 °C in 100 mM Imidazol/MES, pH 6.0, 60 mM calcium/magnesium chloride, 30% PEG 8000/ethylene glycol (w/v) including 10 mM copper(II)chloride dehydrate. Initial crystals were used for macro-seeding to grow larger crystals of the same morphology suitable for X-ray diffraction analysis. Data collection from a single frozen crystal was carried out at beamline 14.2 at the BESSY II electron storage ring (Helmholtz-Zentrum für Materialien und Energie, Berlin; Germany). A 2.2 Å dataset was collected at a wavelength of 0.9184 Å using a CCD detector (MX-225, Rayonics, USA). Diffraction data were processed with the XDS software package[48]. The structure was phased by single wavelength anomalous dispersion (SAD) resulting in the localization of 4 heavy atoms (copper) in space group P4₁ employing the SHELX software suite[49]. Subsequent cycles of heavy atom refinement and density modification, carried out with the software AUTOSHARP[50], yielded an initial electron density which allowed automated tracing of the polypeptide backbone using ARP/wARP from the CCP4 suite[51,52]. The model was completed by manual building using the program Coot[53] and refined with the PHENIX software suite[54]. The structure revealed one Af1 molecule in the asymmetric unit. The electron density was well resolved for most residues of Ub-N and Ub-C, except for the side chains of several evolved residues in the solvent-exposed α2β5 loops (N62, P63, K64, L65, W142, Q143) and the linker region not being visible (A76, G77, I78, G79).

Crystals of the Af2:7B8 complex grew within 1-2 weeks in sitting-drop vapour diffusion plates at 25 °C in 500 mM lithium sulfate, 15% PEG 8000 (w/v). Diffraction data were collected from a single frozen crystal at the BESSY II beamline BL 14.1 using a hybrid pixel detector (Pilatus 6 M, Dectris, Switzerland) and processed with the XDS software package. Phases were determined by Molecular Replacement employing the program PHASER[55] using the three (separated) fibronectin domains of 7B8 (PDB entry: 4GH7[56]) and the two (separated) Ub domains of Af1 as individual search models. Three non-crystallographic symmetry-related complexes were located in the asymmetric unit, each consisting of one Af2 and one 7B8 molecule. The structure was completed using the program Coot[53] and

**Table 2 | Statistics of data collection and structure refinement**

| Dataset (PDB accession) | Af1 (8PF0) | Af2:7B8 complex (8PEQ) |
|---|---|---|
| X-ray source | BESSY BL14.2 | BESSY BL14.1 |
| Wavelength [Å] | 0.9184 | 0.9184 |
| Detector | CCD MX225 | PILATUS 6 M |
| Space group | $P4_1$ | C2 |
| Cell parameter | | |
| a,b,c [Å] | 62.77, 62.77, 67.84 | 168.67, 105.86, 78.99 |
| α,β,γ [°] | 90.00, 90.00, 90.00 | 90.00, 93.52, 90.00 |
| Resolution [Å] | 30.0-2.2 (2.3-2.2) | 43.0-2.3 (2.4-2.3) |
| Completeness [%] | 99.5 (97.6)[a] | 96.8 (78.7)[b] |
| Total reflections unique reflections | 99386 26176 (3189)[a] | 210486 59628 (5777)[b] |
| Multiplicity | 3.8 (3.8)[a] | 3.5 (3.3)[b] |
| $R_{merge}$ | 3.1 (38.7)[a] | 5.8 (39.9)[b] |
| I/σ(I) | 22.9 (3.9)[a] | 16.5 (3.1)[b] |
| $CC_{1/2}$ | 99.9 (91.0)[a] | 99.8 (83.9)[b] |
| Wilson B-factor | 58.6 | 41.9 |
| **Structure refinement** | | |
| Molecules per asymmetric unit | 1 Af1 | 3 Af2 3 7B8 |
| R values [%] | | |
| $R_{work}$ | 19.3 | 19.8 |
| $R_{free}$ | 22.6 | 25.1 |
| Number of atoms | | |
| Protein | 1223 | 9940 |
| $Cu^{2+}$ ions | 4 | – |
| Buffer components | 10 | 10 |
| Solvent | 78 | 536 |
| Average B factor [Å²] | 55.6 | 39.8 |
| Rmsd | | |
| Bond lengths [Å] | 0.007 | 0.008 |
| Bond angles [°] | 1.1 | 1.0 |
| Ramachandran [%] | | |
| Favoured | 98.0 | 97.9 |
| Allowed | 2.0 | 2.0 |
| Outlier | 0.0 | 0.1 |
| Molprobity clashscore | 2.4 | 4.5 |

Values for highest resolution shell are given in parentheses.
[a]Friedel pairs treated as independent reflections.
[b]Friedel pairs merged.

refined with the PHENIX software suite[54]. Structure validation was carried out using MOLPROBITY[57], molecular figures were created with the software PyMOL (Schrödinger LLC, New York, USA). Data collection and refinement statistics are given in Table 2.

**Analytical HPLC size exclusion chromatography**

Analytical size exclusion HPLC of purified complexes of Af1:7B8 and Af2:7B8 were carried out using a Superdex 200 5/150 GL column (GE Healthcare, Freiburg, Germany) equilibrated in PBS at a flow rate of 0.3 ml/min, using a Summit HPLC system (Dionex, Idstein, Germany). The apparent molecular masses were derived from a calibration of the column by linear regression of the retention times of an HPLC gel filtration standard (Bio-Rad, Feldkirchen, Germany) and were compared to theoretical masses of the complexes (47.3 and 47.2 kDa, respectively), calculated from the individual molecular masses (Af1: 17.5 kDa, Af2: 17.4 kDa, 7B8: 29.8 kDa).

**Statistics and reproducibility**

Binding analysis by SPR: $K_D$, $k_{on}$ and $k_{off}$ values were calculated by fitting the traces from five Affilin variant concentrations using a global kinetic fitting (1:1 Langmuir model, BIAevaluation software). Binding analysis by ELISA was conducted in triplicate (to target variants) or by single measurement (to the off-target 6789). IL-2 activity assay: $EC_{50}$ values were calculated from the mean of triplicate measurements using the program SigmaPlot. Data points in corresponding figures show mean values and standard deviations (error bars). Data points of individual measurements were superimposed as coloured crosses (where applicable). Source data of all graphs are available as Supplementary Data 1. Statistics of X-ray data processing and structure refinement are given in Table 2.

**Reporting summary**

Further information on research design is available in the Nature Portfolio Reporting Summary linked to this article.

**Data availability**

The atomic coordinates and structure factors of unbound Af1 and the Af2:7B8 complex have been deposited in the Protein Data Bank, www.pdb.org (PDB ID codes 8PF0 and 8PEQ, respectively).

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

## Acknowledgements

We thank Milton T. Stubbs for helpful discussions, the Helmholtz-Zentrum Berlin für Materialien und Energie for the allocation of synchrotron radiation beam time and the staff of the MX beamlines at the BESSYII storage ring for excellent support.

## Author contributions

A.K., E.F., U.H. and C.P. designed experimental work, A.K. performed biochemical experiments and crystallisation, A.R., F.S. and M.G.-B. carried

out biochemical and cell biological experiments, C.P. solved structures and performed structural analysis, A.K., E.F. and C.P. wrote the manuscript.

## Funding

## Competing interests
A.K., E.F., U.H., F.S., M.G.-B. are employees of Navigo Proteins GmbH. All other authors declare no competing interests. Affilin, Navigo, Navigo Proteins are registered trademarks of Navigo Proteins GmbH.
