## [Peer Review File · Communications Biology]

Reviewers' comments:

Reviewer #1 (Remarks to the Author):

- Does the manuscript have technical or conceptual flaws that should prohibit its publication? No
- Are the conclusions original? Yes,
- Do you feel that the results presented are of immediate relevance for people in your own discipline or for a broader audience? Yes

1. Brief Summary of the Manuscript:

In this study, the authors evolved a di-ubiquitin protein (Affilin) called Af2 to target a specific region of oncofetal fibronectin (EBD). They employed a modified library-based approach, building upon a previously reported protein (Af1, reference 14), to enhance Af2's affinity for the EBD target. The manuscript reports that Af2 and intermediate targets, effectively bind to the EBD with nanomolar affinity. The authors used X-ray crystallography to investigate the binding mode of Af2, revealing atomic details on structural changes responsible for the affinity improvement. The study emphasizes the structural plasticity of the di-ubiquitin scaffold and the significance of structural alterations in enhancing target specificity.

2. Overall Impression of the Work:

The manuscript presents a well-executed research study with a clear and logical narrative structure. The methodology employed is appropriate, and the protein constructs and controls are well-suited for the research objectives. While the research builds upon previous work (Af1), it offers novel structural insights, particularly regarding beta-strand slippage, which is a rare phenomenon but of biological relevance. The study highlights the adaptability and versatility of the ubiquitin scaffold, pointing out the importance of structural modifications in improving target specificity and affinity.

3. General Comments:

I. In the abstract (line 22), the statement challenging “accepted paradigm” that directed evolution of binding proteins should be rooted in rigid frameworks requires clarification. To the best of my knowledge, there is no widely accepted paradigm dictating that protein engineering must exclusively rely on fixed structures. Instead, the limitation lies in the observation that insertions and deletions (InDels) in proteins are significantly more likely to have a deleterious effect than substitutions. This propensity limits their incorporation into randomized protocols. Randomizing a single amino acid position for InDels is complex as two directions can be taken (insertion or deletion) and the modification is not limited to a single residue. However, numerous publications have already emphasized the importance of considering InDels in protein engineering, advocating for their inclusion rather than strict adherence to rigid structural frameworks. For a comprehensive review on InDels and their utilization in protein design, please refer to a recent review by Savino et al. (<https://doi.org/10.1016/j.biotechadv.2022.108010>). In fact, the incorporation of InDels has

been extensively discussed in the literature, and its rational implementation has facilitated the modification of fold architectures, catalysis, and coenzyme binding. Finally, while protein design usually relies on a fixed backbone, alternative methods exist that offer greater flexibility.

II. In the abstract (line 25-26), when stating that insertions and deletions are difficult to predict, consider providing references or examples from the literature to support this claim. Have the authors considered employing computational methods, such as AlphaFold, to substantiate their claim. It would be interesting to compare the predicted structures to the actual crystals structure to see whether they differ.

III. Page 4, lines 81-83. Consider clarifying whether the fine-tuning of the library was instrumental in introducing the observed deletions in Af2 or if these deletions arose randomly during the maturation process. In other words, does the maturation approach applied to Af2 enhance the likelihood of incorporating more InDels when compared to the Af1 approach?

IV. Consider explain the rationale behind the selected seven mutations more explicitly, considering that some readers may not be familiar with the previous publication describing the maturation of the Af1 variant.

V. Page 4, line 126. Consider briefly explaining the nomenclature for beta-strand slippage to make it more accessible to a broader audience, such as mentioning that a negative number indicates a shift towards the C-terminus.

VI. Suggest changing the term "structurally inert scaffold model" on page 6, line 208, to "fixed backbone design" to align with common nomenclature.

VII. In the second and third paragraphs on page 6, consider adding the example of beta slippage described by Eneqvist et al., which also involves the rearrangement of leucine residues, to the list of examples discussed in the manuscript (lines 217-244). (<https://www.sciencedirect.com/science/article/pii/S1097276500001179?via=ihub>).

VIII. In the last paragraph on page 6 (lines 245-247), the beta-slippage presented here is the key finding of this work and certainly an interesting concept that emphasizes the importance of insertions and deletions (InDels) in protein engineering. However, it's important to clarify that this phenomenon may not necessarily be a universal characteristic of all protein folds. Nevertheless, the di-ubiquitin scaffold serves as an excellent model for illustrating this concept, as slippage appears to be an inherent feature of this particular scaffold.

Reviewer #2 (Remarks to the Author):

This manuscript describes the selection, from a combinatorial library, of an Affilin protein (Af2), comprising a fusion of two copies of mutated ubiquitin, as an artificial binding protein directed against the extra-domain B (EDB) of oncofetal fibronectin, an established marker of tumor neovasculature. X-ray structural analysis of the complex between Af2 and the Fn7B8 fragment revealed unexpected register shifts of the β 5 strand in each of the two ubiquitin modules, by two and four residues, respectively, leading to a newly reshaped target-binding interface around the neighboring loop region.

This work builds upon an earlier publication from the same group where a related diubiquitin-based Affilin (Af1) with evolved binding properties against the same oncofetal Fn target had been described (Ref. 14). Its crystal structure in the absence of the target was solved here, too, indicating a similar β 5 strand slippage. As such, the findings are of interest to an audience in the areas of structural biology and protein engineering. In the absence of animal experiments, the functional data presented for the EDB-specific Affilin-IL-2 fusion protein are insufficient to assess any perspective for medical applications. Hence, this manuscript appears better suited to a more specialized journal.

The authors try to present their findings in a broader sense that "fold plasticity allowing β -strand slippages enhances the evolutionary potential of proteins beyond "simple" mutations significantly and provides a general mechanism to generate residue insertions/deletions in proteins." However, this view appears exaggerated as there are published examples of other alternative binding proteins with an unexpected change in protein fold or mode of target recognition, see e.g. PMIDs 18375754 and 21968397. Furthermore, the described β -strand slippage is a phenomenon well known for natural ubiquitin, as the authors have already discussed (see p. 6 of the main manuscript). In fact, from the combinatorial protein design perspective, the simple register shift by two or four residues, allowed by the alternating spacing of a series of Leu residues on the β -strand, just leads to a three-fold larger structure/sequence space, which is less than the effect of a single additional amino acid residue that would get randomized at a critical position.

Finally, before publication can be recommended the entire manuscript needs to be reorganized. While the data themselves seem solid, their way of presentation appears more than confusing. In the Results section there is way too much reference to the Supplementary Information, which not only includes a large number of Figures but also an extended "Supplementary discussion". On top of that, the main manuscript contains five "Extended data figures", each with multiple panels. Together, this appears overloaded and the authors should do a better job to focus their message. On the other hand, the structural presentations in Figure 1d,e and Figure 2 of the main manuscript should be improved to more clearly illustrate the findings described in the text.

Reviewer #3 (Remarks to the Author):

The paper from Katzschmann et al. describes the directed evolution of a di-ubiquitin scaffold (termed Affilin) towards high affinity and specific binding to oncofetal fibronectin. This work

expands on the 2014 paper from Lorey et al., which used a similar approach to the generation of high affinity binding protein from di-ubiquitin scaffolds. Beyond the generation of a low nanomolar and specific binder to the extra-domain B (EDB) of oncofetal fibronectin, the main finding of the paper is the binding mode adopted by the evolved Affilin molecule, which the authors reveal through structural studies, and shows an unexpected register shift in one of the strands when compared to the canonical ubiquitin structure. The author points to this structural plasticity as a valuable element to consider when evolving protein scaffolds.

Overall the study was well-conducted, the data presented clearly, and the findings interesting, especially for readers interested in the directed evolution of artificial binding proteins for biotechnological applications. Below are my comments (general and specific), which I think would enhance the readability of the paper for a broader audience

— General comments —

The first result section (“Generation of Affilin molecules targeting EDB of oncofetal fibronectin”) would benefit from being expanded, and some details more clearly explained. While some of these can be found in the Supplementary discussion, they should be moved to the main text. In particular:

Choice of positions for saturation mutagenesis during PD? The rationale is only partially explained in the SI.

Size of the libraries (both PD and RD).

The reason for first doing PD, followed by RD? It is very unclear to this reviewer why the author chose this switch, and the reader would benefit from understanding the rationale being the choice of methods

The second result section (“Target-bound Af2 reveals distinct register shifts in the $\beta 5$ strands”) feels a bit lengthy and too detailed. While the discussion of the register shift is important given the conclusion of the paper, it should be shortened for readability.

The third result section (“The structure of Af1 reveals $\beta 5$ register-shifts in absence of the target”) also feels a bit lengthy, especially given that it describes the structure of a molecule that was the subject of the 2014 paper by Lorey et al. A similar point (the fact that the register shift exists in the absence of the target) could be made more directly. At the end the authors speculate about the binding mode of Af1 compared to Af2; perhaps the authors could include an AlphaFold2 prediction of Af1 bound to its target and compare it to the crystal structure of Af2 bound to 7B8?

Af2 does not bind 67B, yet this target has interfaces I+II (Fig. 1D), which the authors show are major contributors to the overall binding of Af2 to (6)7B8(9) (e.g. molecular surface area). The authors should speculate as to why this is the case, as it appears odd that no binding was observed at all? Similar question for 6789 since this target has interface III, which although it is smaller than I+II, could still potentially lead to some weak binding?

Interface III contains no evolved residues but provides binding to Fn8 via structural rearrangements of its ‘native’ residues due to the register shift. This should be more clearly explained/highlighted in

the main text as it is an interesting finding, and further substantiate the author's claim that structural plasticity of artificial binding scaffolds may have been under-appreciated to date.

— Specific comments —

A table with all the amino acid sequences used in the study should be provided (ideally highlighting any tags/modifications compared to the WT sequences, e.g. for the different Fn constructs, but also for all the Af molecules).

Fig 1B; the x-axis should be on a log-scale in order to be able to properly assess the goodness of the fit (same comment applies to extended Fig. 2).

Fig. 1C; the description of the second cell line (NHDf) is missing in the legend.

Fig. S2; the flow rate should be indicated in the figure legend, or the SEC x-axis changed to retention volume.

The RMSD between the different complexes of the asymmetric unit should be indicated in the main text, ideally for the full complex and for Af2 alone in order to provide the reader with a more quantitative picture of the structural deviations observed.

By how many mutations do Af1 and Af2 differ? This should be indicated in the main result section since the register shift of Af1 in the absence of target is used as a proxy for the behavior of Af2.

Fig. 3C; the EC50 would be better presented as two overlaid titration curves in order for the reader to assess the results. The two bar plots do not provide any additional information compared to just quoting the numbers in the text.

Mutations Af2s-GL, Af2s-AL, and Af2s-deltaDP-KS are only described in the SI, not the main text, but feature in Table 1. This should be changed (either by removing these entries from the main text Table, or by describing these mutations in the main text).

Extended Fig. 5; it would be interesting to see an overlay of just the Ub-N domain of Af1 and Af2, and another overlay of just the Ub-C of Af1 and Af2 to see how these Affilin molecules differ from one another beyond their relative orientation currently shown in panel b.

The Editorial Office
c/o Dr Abriata

Manuscript submission to Communications Biology
COMMSBIO-23-3005-A

April 16th 2024

Point-by-point replies to referees' comments

Thank you very much for your comments on our manuscript titled 'Ubiquitin-derived artificial binding proteins targeting oncofetal fibronectin reveal scaffold plasticity by β -strand slippage'. They have been very helpful in the revision process. We have addressed the concerns raised to improve the quality and clarity of our manuscript. The changes made to the manuscript are outlined in the numbered tables below, with quotes from the referees' comments in italics and our responses in standard text underneath.

Reviewer #1	
1.1	In the abstract (line 22), the statement challenging “accepted paradigm” that directed evolution of binding proteins should be rooted in rigid frameworks requires clarification. To the best of my knowledge, there is no widely accepted paradigm dictating that protein engineering must exclusively rely on fixed structures. Instead, the limitation lies in the observation that insertions and deletions (InDels) in proteins are significantly more likely to have a deleterious effect than substitutions. This propensity limits their incorporation into randomized protocols. Randomizing a single amino acid position for InDels is complex as two directions can be taken (insertion or deletion) and the modification is not limited to a single residue. However, numerous publications have already emphasized the importance of considering InDels in protein engineering, advocating for their inclusion rather than strict adherence to rigid structural frameworks. For a comprehensive review on InDels and their utilization in protein design, please refer to a recent review by Savino et al. (https://doi.org/10.1016/j.biotechadv.2022.108010). In fact, the incorporation of InDels has been extensively discussed in the literature, and its rational implementation has facilitated the modification of fold architectures, catalysis, and coenzyme binding. Finally, while protein design usually relies on a fixed backbone, alternative methods exist that offer greater flexibility. The corresponding section of the abstract text has been re-phrased, removing the “accepted paradigm” statement, now focusing more on the implications of β-strand register shifts for protein evolution. Revised text: Abstract (page 2, lines 22-26): “Protein backbone alterations resulting from β-strand register shifts, as seen in the ubiquitin fold, can pose additional challenges to protein engineering as structural evidence of these events is still limited and they are difficult to predict. However, they can surface under the selection pressure of directed evolution and suggest that backbone plasticity allowing β-strand

	slippages can increase structural diversity, enhancing the evolutionary potential of a protein scaffold.” We also refer to the particular challenges imposed by the implementation of ‘InDels’ (insertions and deletions) in protein engineering approaches in the introduction and in the discussion. Added text: Introduction (page 2, lines 42-43): “Incorporating residue insertions or deletions (InDels) into evolutionary strategies introduces additional complexity, as they may result in larger, non-local structural changes (Savino et al).” Discussion (page 7, lines 288-296): “Register shifts of β-strands may provide a conceptual framework for the evolution of sequence insertions and deletions (InDels) at the protein structure level to generate stable neofunctional domains. They add a fascinating aspect to (directed) protein evolution beyond localized mutations, as these events place evolved residues into new positions and recruit neighbouring amino acids into new environments. However, designing InDels rationally remains a complex task, despite significant progress in implementing them into protein engineering approaches (Savino et al). Protein backbone plasticity that allows for the generation of InDels, for instance through β-strand slippage, might help simplifying this task. These events can enhance the structural diversity of the candidate pool without the need to expand the size of the genetic library.” Deletions that occurred during the evolution of the Affilin variants were not intentionally incorporated in our approach; rather, they occurred randomly. They compensate (and stabilise) the β-strand register shifts that were crucial in conferring binding properties to the Affilins. The “insertions” extending the $\alpha 2\beta 5$ loops have not been introduced directly by genetic insertion, they are a structural consequence of mutations leading to the register shifts. Thus, our findings contribute to the topic of InDels by demonstrating that they can also result from scaffold plasticity, which allows for β-strand slippage.
1.2	In the abstract (line 25-26), when stating that insertions and deletions are difficult to predict, consider providing references or examples from the literature to support this claim. Have the authors considered employing computational methods, such as AlphaFold, to substantiate their claim. It would be interesting to compare the predicted structures to the actual crystals structure to see whether they differ. We’ve incorporated the results of AlphaFold predictions for Af1 and Af2, respectively (Supplementary Figure S13). They demonstrate Alphafold's current capabilities do not allow for the prediction of the β-strand register shifts observed in the crystal structures of Af1 and Af2. The following sentences were added to the manuscript text: Results (page 5, lines 192-195): “Our attempts to predict the $\beta 5$ register shifts of Af1 and Af2 using Alphafold ¹⁹ have been unsuccessful, both in the absence and presence of the target 7B8. In none of the predicted structures register shifts of $\beta 5$ were observed, precluding the prediction of a potential Af1 complex structure (Supplementary Figure S13).” Discussion (page 7, lines 254-257): “These backbone rearrangements are not easily predictable from the sequence and may even be missed in (lower resolution) experimental structures. Recent structure prediction

	algorithms, including Alphafold, are currently unable to predict the β -strand register shifts observed in the Affilin crystal structures, as our attempts have shown.”
1.3	Page 4, lines 81-83. Consider clarifying whether the fine-tuning of the library was instrumental in introducing the observed deletions in Af2 or if these deletions arose randomly during the maturation process. In other words, does the maturation approach applied to Af2 enhance the likelihood of incorporating more InDels when compared to the Af1 approach? The deletions occurred randomly. The library was not fine-tuned and the maturation strategy was not designed to incorporate deletions (see comment 1.1). We have re-phrased the corresponding parts in the text to clarify. Results (page 3, lines 80-81): “For affinity maturation of Af2p, we opted for a broader mutagenesis approach by error-prone PCR, allowing re-randomisation of amino acids at any position.” Results (page 3, lines 89-91): “Af2s differed from Af2p by having two additional mutations (Pro38Gln and Tyr143Phe) and a randomly arisen deletion (Ile78) in the linker region between the N- and C-terminal Ub domains.”
1.4	Consider explain the rationale behind the selected seven mutations more explicitly, considering that some readers may not be familiar with the previous publication describing the maturation of the Af1 variant. The corresponding chapter was moved from the Supplementary Discussion to the main text and extended with additional details. Results (page 3, lines 62-70) “Using in silico analysis, potential binding epitopes of the Ub scaffold with a high tolerance for amino acid substitutions were evaluated by assessing protein stability perturbations induced by single amino acid exchanges in wild-type Ub. Nine amino acid positions (2, 4, 6, 8, 62-66, positions 2-8 located in strand β1 and loop β1β2, positions 62-66 in loop α2β5 and strand β5) were selected for randomisation, from which subsets of 8 and 6 positions were used for saturation mutagenesis of the two Ub domains (sequence overview given in Supplementary Figure S1). We followed a similar approach to generate the Affilin Af2, employing the diubiquitin scaffold with a subset of 7 of the same randomised positions in each Ub domain, a modified library design and advanced selection and maturation procedures.” Related information can also be found in the method section (page 8, lines 337-341).
1.5	Page 4, line 126. Consider briefly explaining the nomenclature for beta-strand slippage to make it more accessible to a broader audience, such as mentioning that a negative number indicates a shift towards the C-terminus. We have revised the text to clarify, that negative register shifts correspond to relocation of residues towards the N-terminus. Results (page 4, lines 138-142): “The most striking observation was made by the discovery that the β5-strands of both Ub domains of target-bound Af2 have slipped from their original position in the β-sheet (as in the structure of Ub wt) by several residues towards the N-terminus (Figure 3). The resulting negative register shifts of β5 were clearly visible in the $2F_o - F_c$ electron density of the Af2:7B8 complex, (Supplementary Figure S8a, b) and are present in all complexes of the asymmetric unit.”

1.6	Suggest changing the term "structurally inert scaffold model" on page 6, line 208, to "fixed backbone design" to align with common nomenclature.
	We removed this phrase from this sentence, as it is not required for our statement given (discussion, page 6, lines 235-236).
1.7	In the second and third paragraphs on page 6, consider adding the example of beta slippage described by Eneqvist et al., which also involves the rearrangement of leucine residues, to the list of examples discussed in the manuscript (lines 217-244). (https://www.sciencedirect.com/science/article/pii/S1097276500001179?via=ihub).
	A reference has been added to the discussion. Discussion (page 7, lines 272-275) "The structural changes induced by a three-residue negative register shift of a β -strand in human transthyretin, a tetrameric plasma protein associated with amyloidosis, are directly linked to amyloid fibril formation (Eneqvist et al.)."
1.8	In the last paragraph on page 6 (lines 245-247), the beta-slippage presented here is the key finding of this work and certainly an interesting concept that emphasizes the importance of insertions and deletions (InDels) in protein engineering. However, it's important to clarify that this phenomenon may not necessarily be a universal characteristic of all protein folds. Nevertheless, the di-ubiquitin scaffold serves as an excellent model for illustrating this concept, as slippage appears to be an inherent feature of this particular scaffold.
	The following parts of the discussion have been rephrased to account for this concern and discuss the perspectives of register-shift induced InDels for protein evolution. Discussion (page 7, lines 280-281): " β -strand register shifts represent one facet of a more general plasticity of β -sheets, although this may not surface for every fold containing β -sheets." Discussion (page 7, lines 288-298): "Register shifts of β -strands may provide a conceptual framework for the evolution of sequence insertions and deletions (InDels) at the protein structure level to generate stable neofunctional domains. They add a fascinating aspect to (directed) protein evolution beyond localized mutations, as these events place evolved residues into new positions and recruit neighbouring amino acids into new environments. However, designing InDels rationally remains a complex task, despite significant progress in implementing them into protein engineering approaches (Savino et al). Protein backbone plasticity that allows for the generation of InDels, for instance through β -strand slippage, might help simplifying this task. These events can enhance the structural diversity of the candidate pool without the need to expand the size of the genetic library. Although speculative at this point, in the case of the Affilin design, the two tethered Ub domains could potentially exhibit three different register-shifted states (0, 2, -4) each, resulting in a nine-fold increase in the number of structural variants in total."

Reviewer #2	
2.1	This work builds upon an earlier publication from the same group where a related diubiquitin-based Affilin (Af1) with evolved binding properties against the same oncofetal Fn target had been described (Ref. 14). Its crystal structure in the absence of the target was solved here, too, indicating a similar beta5 strand slippage. As such, the findings are of interest to an audience in the areas of structural biology and protein engineering. In the absence of animal experiments, the functional data presented for the EDB-specific Affilin-IL-2 fusion protein are

	insufficient to assess any perspective for medical applications. Hence, this manuscript appears better suited to a more specialized journal. In our opinion, data from animal experiments for the Affilin-IL-2 fusion would certainly be significant for future applied aspects of our work but are beyond the scope of the present report. This manuscript focuses on the structural and functional implications of β-strand slippages observed in the Affilin scaffold. However, we believe these findings are biologically relevant to protein evolution in general and may be of interest to a wider audience. To lower the tone in the chapters regarding potential application, the corresponding parts have been re-phrased to indicate that this aspect of our work is still at the 'proof-of-concept' stage. Introduction (page 1, lines 57-58): “Moreover, with a view to potential future applications, we show that a genetic Affilin-cytokine fusion is functional and able to target EDB-expressing cells in vitro.” Results (page 6, line 222-224): “This demonstrates that the chimeric Affilin-cytokine fusion is capable of targeting EDB-expressing cells in vitro and that the activity of the payload IL-2 is not significantly affected in context of Af2.” Discussion (page 8, lines 309-311): “Finally, combining the de-novo binding properties of an evolved Affilin with the function of a payload protein, as demonstrated for the Affilin-IL2 chimera, offers new perspectives towards their use in medical and biotechnological applications.”
2.2	The authors try to present their findings in a broader sense that "fold plasticity allowing β-strand slippages enhances the evolutionary potential of proteins beyond "simple" mutations significantly and provides a general mechanism to generate residue insertions/deletions in proteins." However, this view appears exaggerated as there are published examples of other alternative binding proteins with an unexpected change in protein fold or mode of target recognition, see e.g. PMIDs 18375754 and 21968397. Furthermore, the described β-strand slippage is a phenomenon well know for natural ubiquitin, as the authors have already discussed (see p. 6 of the main manuscript). In fact, from the combinatorial protein design perspective, the simple register shift by two or four residues, allowed by the alternating spacing of a series of Leu residues on the β-strand, just leads to a three-fold larger structure/sequence space, which is less than the effect of a single additional amino acid residue that would get randomized at a critical position. Although the phenomenon of β-strand slippage has been reported previously, we believe the Affilin structures represent outstanding examples of how protein evolution exploits fold plasticity through β-strand slippage. The achieved backbone rearrangements mimic the consequences of the well-studied InDels (genetic insertions/deletions), but are achieved on a structural level, based on the plasticity of the ubiquitin scaffold (stabilized by the evolved amino acids in Af1 and Af2). As discussed in our article, such events are not limited to the ubiquitin scaffold but occur naturally also in other proteins (although rarely observed and presumably under-represented in structural databases). Implications of β-strand register shifts for combinatorial protein design: a scaffold allowing three different register-shifted states (e.g. 0,-2,-4) increases the structural diversity of a derived candidate protein pool by three-fold, as outlined by reviewer #2. However, the Affilin design comprising two tethered Ub domains multiplies the potential structural space to a nine-fold increase. Notably, this gain of diversity comes at no additional cost, as it does not require expansion of the genetic library.

	However, we understand the raised concerns and have revised our statements in the abstract and in the discussion accordingly. The recommended citations were added to the manuscript, exemplifying other cases of unexpected structural changes, arising from the evolution of artificial binding proteins. Abstract (page 2, lines 21-26): “Protein backbone alterations resulting from β-strand register shifts, as seen in the ubiquitin fold, can pose additional challenges to protein engineering as structural evidence of these events is still limited and they are difficult to predict. However, they can surface under the selection pressure of directed evolution and suggest that backbone plasticity allowing β-strand slippages can increase structural diversity, enhancing the evolutionary potential of a protein scaffold.” Discussion (pages 7-8, lines 288-307): “Register shifts of β-strands may provide a conceptual framework for the evolution of sequence insertions and deletions (InDels) at the protein structure level to generate stable neofunctional domains. They add a fascinating aspect to (directed) protein evolution beyond localized mutations, as these events place evolved residues into new positions and recruit neighbouring amino acids into new environments. However, designing InDels rationally remains a complex task, despite significant progress in implementing them into protein engineering approaches (Savino et al). Protein backbone plasticity that allows for the generation of InDels, for instance through β-strand slippage, might help simplifying this task. These events can enhance the structural diversity of the candidate pool without the need to expand the size of the genetic library. Although speculative at this point, in the case of the Affilin design, the two tethered Ub domains could potentially exhibit three different register-shifted states (0, 2, -4) each, resulting in a nine-fold increase in the number of structural variants in total. The structure determination of the oncofetal-fibronectin-specific Affilin variants Af1 and Af2 presented here, demonstrate that scaffold plasticity was crucial in obtaining high affinity binders, as β-strand register shifts have been observed in all four Ub domains. They have co-evolved with amino acid composition under the selection pressure of directed evolution. The structures also exemplify potential caveats in the interpretation of results from combinatorial protein engineering and underline the importance of structural analyses. As demonstrated for the Affilin molecules, such endeavours can uncover unforeseeable structural alterations. Our data add to previous reports on artificially evolved binding proteins, which describe for instance unexpected binding modes for protein Z-based affibodies (Hoyer et al.) and a single-domain antibody fragment (Schiefner et al.).”
2.3	Finally, before publication can be recommended the entire manuscript needs to be reorganized. While the data themselves seem solid, their way of presentation appears more than confusing. In the Results section there is way too much reference to the Supplementary Information, which not only includes a large number of Figures but also an extended "Supplementary discussion". On top of that, the main manuscript contains five "Extended data figures", each with multiple panels. Together, this appears overloaded and the authors should do a better job to focus their message. On the other hand, the structural presentations in Figure 1d,e and Figure 2 of the main manuscript should be improved to more clearly illustrate the findings described in the text. The manuscript has undergone substantial reorganisation. The revised version comprises now five re-assembled, reworked figures in the main article (no extended figures). Figure 4 contains additional representations requested by reviewer #3. For clarity, and to avoid excessive referencing to the Supplementary Information (SI), Figures 1b, 2d and 4c moved from the SI to

	the main article. Other (extended) figures moved to the SI. Revised table 1 contains only the Affilin variants mentioned in the main text. Table 2 has moved from the SI to the main article to comply with the submission guide. The Supplementary Discussion was significantly reduced and contains only the discussion of Affilin variants assessing the effects of affinity maturation and the description of the molecular interactions underlying stabilisation of the different register shifts and target binding.
--	---

Reviewer #3	
3.1	The first result section (“Generation of Affilin molecules targeting EDB of oncofetal fibronectin”) would benefit from being expanded, and some details more clearly explained. While some of these can be found in the Supplementary discussion, they should be moved to the main text. In particular: Choice of positions for saturation mutagenesis during PD? The rationale is only partially explained in the SI. Size of the libraries (both PD and RD). The chapter addressing the first point was moved from the Supplementary Discussion to the main text and extended with additional details. Results (page 3, lines 62-70) “Using in silico analysis, potential binding epitopes of the Ub scaffold with a high tolerance for amino acid substitutions were evaluated by assessing protein stability perturbations induced by single amino acid exchanges in wild-type Ub. Nine amino acid positions (2, 4, 6, 8, 62-66, positions 2-8 located in strand β1 and loop β1β2, positions 62-66 in loop α2β5 and strand β5) were selected for randomisation, from which subsets of 8 and 6 positions were used for saturation mutagenesis of the two Ub domains (sequence overview given in Supplementary Figure S1). We followed a similar approach to generate the Affilin Af2, employing the diubiquitin scaffold with a subset of 7 of the same randomised positions in each Ub domain, a modified library design and advanced selection and maturation procedures.” Related information can also be found in the method section (page 8, lines 337-341). Information on the library sizes was added to the main text. Results (page 3, line 73): “The effective library size used in PD was calculated to be $2.5 \cdot 10^9$.” Results (page 3, lines 81-83): “Selection of molecules binding to the target 67B89 was performed using four rounds of ribosome display (RD), taking into account a theoretical library size of $6 \cdot 10^{10}$ variants, which is significantly larger compared to the library used in PD.”
3.2	The reason for first doing PD, followed by RD? It is very unclear to this reviewer why the author chose this switch, and the reader would benefit from understanding the rationale being the choice of methods Error-prone PCR allowed re-randomization of amino acids at all positions during affinity maturation, resulting in a potentially significantly larger library size (see comment 3.1). To account for this, selection by RD was chosen which can map larger libraries than PD. In our efforts, switching the selection strategy has led to more consistent results at initial stages of

	Affilin development (data not shown) and could circumvent potential limitations inherent to PD. We re-phrased the following sentences: Results (page 3, lines 80-83): “For affinity maturation of Af2p, we opted for a broader mutagenesis approach by error-prone PCR, allowing re-randomisation of amino acids at any position. Selection of molecules binding to the target 67B89 was performed using four rounds of ribosome display (RD), taking into account a theoretical library size of $6 \cdot 10^{10}$ variants, which is significantly larger compared to the library used in PD.”
3.3	The second result section (“Target-bound Af2 reveals distinct register shifts in the β5 strands”) feels a bit lengthy and too detailed. While the discussion of the register shift is important given the conclusion of the paper, it should be shortened for readability. The result section (renamed to “Target binding involves different register shifts in the β5 strands of Af2”) has been revised to address this concern (page 4 and 4). Although the text discussing the register shifts has been condensed, the length of the complete section did not decrease, as other comments from the reviewer required adding more text (e.g. description of the three complexes in the asymmetric unit, previously located in the Supplementary Information, see below).
3.4	The third result section (“The structure of Af1 reveals β5 register-shifts in absence of the target”) also feels a bit lengthy, especially given that it describes the structure of a molecule that was the subject of the 2014 paper by Lorey et al. A similar point (the fact that the register shift exists in the absence of the target) could be made more directly. At the end the authors speculate about the binding mode of Af1 compared to Af2; perhaps the authors could include an AlphaFold2 prediction of Af1 bound to its target and compare it to the crystal structure of Af2 bound to 7B8? The text describing the register shifts in Af1 has been reduced for clarity. However, including a chapter discussing how the sequence of Af1 and Af2 differ, as requested by the reviewer in a comment below, required additional descriptions in this result section. We also added, that AlphaFold is currently not able to predict the register shifts observed in Af1 and Af2, respectively (see Supplementary Figure 13). As a test, we also attempted to predict the structure of the Af2:7B8 complex using AlphaFold, which resulted in hypothetical complex structures that suggested completely different binding modes compared to the crystal structure (data not shown). This precluded the prediction of a putative Af1:7B8 complex structure using AlphaFold. The corresponding text additions are: Results (page 5, lines 192-195): “Our attempts to predict the β5 register shifts of Af1 and Af2 using AlphaFold have been unsuccessful, both in the absence and presence of the target 7B8. In none of the predicted structures register shifts of β5 were observed, precluding the prediction of a potential Af1 complex structure (Supplementary Figure S13).” Discussion (page 7, lines 254-257): “These backbone rearrangements are not easily predictable from the sequence and may even be missed in (lower resolution) experimental structures. Recent structure prediction algorithms, including AlphaFold, are currently unable to predict the β-strand register shifts observed in the Affilin crystal structures, as our attempts have shown.”
3.5	Af2 does not bind 67B, yet this target has interfaces I+II (Fig. 1D), which the authors show are major contributors to the overall binding of Af2 to (6)7B8(9) (e.g. molecular surface area). The authors should speculate as to why this is the case, as it appears odd that no binding was

	observed at all? Similar question for 6789 since this target has interface III, which although it is smaller than I+II, could still potentially lead to some weak binding? The following text was added to the discussion to address this comment: Discussion (page 6, lines 243-253): “The binding mode elucidated from the Af2:7B8 complex structure can rationalise the observations from the binding analysis (Figures 1 and 2). Af2 binds to the Fn variants 7B8, 67B89 and B89 with high affinity through binding interfaces I to III that involve solely the EDB and Fn8. In contrast, Af2 does not exhibit significant affinity to 67B, despite the theoretical possibility of binding the EDB through interfaces I and II. However, in addition to EDB, Fn8 is also crucial for the interaction, indicating that the engagement of interface III is necessary for either the correct alignment of the Ub-N and Ub-C domains or the structural stabilisation of the -4 register shift in target-bound Ub-C, or both. Likewise, the absence of the EDB and therefore interfaces I and II, which provide the major molecular surface area (853 Å²) for binding, can explain the lack of Af2 binding to 6789. Although Fn8 could theoretically engage interface III (326 Å²), it is conceivable that in the absence of the EDB, a -4 register shift (stabilised by target binding) may not be present, rendering the domain Ub-C binding-incompetent.”
3.6	Interface III contains no evolved residues but provides binding to Fn8 via structural rearrangements of its ‘native’ residues due to the register shift. This should be more clearly explained/highlighted in the main text as it is an interesting finding, and further substantiate the author’s claim that structural plasticity of artificial binding scaffolds may have been under-appreciated to date. We have re-phrased the corresponding text to make this point more clear. Results (page 4, lines 147-149): “Interestingly, only non-evolved Af2 residues successive to β5 contribute to binding interface III. These amino acids have been relocated by the register shifts into new positions where they contribute to target binding (Figure 2d).”
3.7	A table with all the amino acid sequences used in the study should be provided (ideally highlighting any tags/modifications compared to the WT sequences, e.g. for the different Fn constructs, but also for all the Af molecules). The amino acid sequences of the Affilin variants and the used targets (fibronectin fragments), including tag information, are given in the Supplementary Figures S1 and S2.
3.8	Fig 1B; the x-axis should be on a log-scale in order to be able to properly assess the goodness of the fit (same comment applies to extended Fig. 2). The Figures 1c and 5a have been revised accordingly.
3.9	Fig. 1C; the description of the second cell line (NHDF) is missing in the legend. The missing description was added to the legend of (revised) Figure 1d. “Binding of Af2s to human fetal lung fibroblasts cells (Wi-38, high EBD expression, detected by immunofluorescence in green, cell nuclei stained in blue) and no binding of Af2s to neonatal human dermal fibroblasts (NHDF, no EDB expression).”
3.10	Fig. S2; the flow rate should be indicated in the figure legend, or the SEC x-axis changed to retention volume. The flow rate (0.3 ml/min) has been added to the legend of Supplementary Figure S4.
3.11	The RMSD between the different complexes of the asymmetric unit should be indicated in the main text, ideally for the full complex and for Af2 alone in order to provide the reader with a more quantitative picture of the structural deviations observed. The RMSD have been included in the main text (previously in the SI).

	Results (page 4, lines 117-122): “The superposition of the three complexes reveal a high overall structural similarity, with an average Cα RMSD of 3.7 Å. The Af2 molecules (chains L, J, M) show less structural differences with an average Cα RMSD of 2.0 Å, compared to the 7B8 molecules (chains A, B, C) with an average CαRMSD of 4.3 Å. The largest deviations result from the shifted position of Fn7 in the C:M complex, as indicated by an average Cα RMSD of 7.0 Å to Fn7 in the A:L and B:J complexes.”
3.12	By how many mutations do Af1 and Af2 differ? This should be indicated in the main result section since the register shift of Af1 in the absence of target is used as a proxy for the behavior of Af2. We have included the requested information in the result section. Results (page 5, lines 170-175): “Af1 and Af2 were independently evolved against the same target 67B89. Four of the substituted Af1 residues are identical in Af2, including H6, P63, L65 and Q86. Both Affilin variants differ at 14 positions, including three unique positions that were randomised only in Af1 (R2, W4) and Af2 (D8), respectively. The deletion at position 140 of Af2 (ΔQ140) was not observed in Af1 and two changes were uniquely evolved in Af2 during the affinity maturation (Q38, ΔI78 in the linker, full sequences of Af1 and Af2 given in Supplementary Figure S1).”
3.13	Fig. 3C; the EC50 would be better presented as two overlaid titration curves in order for the reader to assess the results. The two bar plots do not provide any additional information compared to just quoting the numbers in the text. Figure 5c has been revised accordingly. For technical reasons this required re-evaluation of the raw data, resulting in slightly different EC₅₀ values. The corresponding chapter in the result section has been updated. Results (page 6, lines 221-222): “The comparison of mean EC₅₀ values showed similar potencies of refolded Af2-IL-2 (EC₅₀= 51 \pm 4 pM) and recombinant IL-2 (EC₅₀=46 \pm 5 pM)(Figure 5c).”
3.14	Mutations Af2s-GL, Af2s-AL, and Af2s-deltaDP-KS are only described in the SI, not the main text, but feature in Table 1. This should be changed (either by removing these entries from the main text Table, or by describing these mutations in the main text). The corresponding entries were moved from Table 1 to Supplementary Table 1.
3.15	Extended Fig. 5; it would be interesting to see an overlay of just the Ub-N domain of Af1 and Af2, and another overlay of just the Ub-C of Af1 and Af2 to see how these Affilin molecules differ from one another beyond their relative orientation currently shown in panel b. The requested overlays were added to Figure 4 (d,e) and the following text was included in the result section: Results (page 5, lines 183-190): “The molecular interactions stabilising the -2 register shifts of Af1 are similar to those in Af2 Ub-N (described in detail in the Supplementary Discussion). The very similar structures of the Ub-N domains of Af1 and Af2 (Cα RMSD 1.5 Å) both display a -2 register shift in β5. However, there are differences in the conformations of the α2β5 loops (Figure 4d). The structural superposition of the Ub-C domains of Af1 and Af2 (Cα RMSD 2.5 Å) illustrates the plasticity of the Affilin scaffold. The individual register shifts (-2/-4) remodel the α2β5 loop differently, depending on the number of accommodated residues and results in corresponding retractions of the C-terminal residues towards the target binding site (Figure 4e).”

A more detailed analysis of the molecular interactions stabilizing the Ub domains of Af1 and Af2 is given in the Supplementary Discussion.

Implications for a putatively similar target recognition mode of Af1 Ub-N are also discussed in this section (Results, page 5 and 6, lines 198-206, Supplementary Figure S14).

Yours sincerely,

Christoph Parthier

REVIEWERS' COMMENTS:

Reviewer #1 (Remarks to the Author):

Having now read through the comments, I consider that the authors have adequately addressed the concerns raised. The revisions provided additional clarity, and the missing points have been sufficiently resolved.

Reviewer #2 (Remarks to the Author):

The authors have done a good job in the revision of their manuscript, which has been considerably improved compared with the original version.

Just two minor points:

- line 36: Please correct "ankryrin repeat proteins" to "ankyrin repeat proteins".
- page 4: It seems that references to Supplementary figures S6 and S7 are missing here, please amend.